# Segment Everything Everywhere All at Once

**Xueyan Zou**[*§2], **Jianwei Yang**[*‡1], **Hao Zhang**[*♯], **Feng Li**[*♯], **Linjie Li**[†], **Jianfeng Wang**[†]
**Lijuan Wang**[†], **Jianfeng Gao**[¶‡], **Yong Jae Lee**[¶§]

[§] University of Wisconsin-Madison   [‡] Microsoft Research, Redmond   [♯] HKUST   [†] Microsoft Cloud & AI

[*]Equal Contribution   [¶] Equal Advisory Contribution   1. Project Lead   2. Main Technical Contribution

{xueyan,yongjaelee}@cs.wisc.edu   {jianwyan,jfgao,linjli}@microsoft.com   {hzhangcx,fliay}@connect.ust.hk

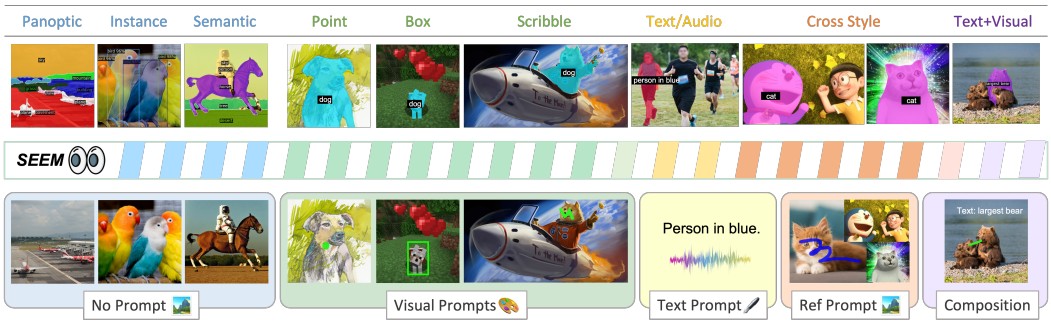

Figure 1: *SEEM* supports generic segmentation tasks—including semantic, instance, and panoptic segmentation—in an open-set fashion when no prompt is provided. *SEEM* also enables the use of visual, textual, and referring region prompts in flexbile combinations, making it a promptable and interactive segmentation interface.

## Abstract

In this work, we present ***SEEM***, a promptable and interactive model for segmenting everything everywhere all at once in an image, as shown in Fig. 1. In *SEEM*, we propose a novel decoding mechanism that enables diverse prompting for all types of segmentation tasks, aiming at a universal segmentation interface that behaves like large language models (LLMs). More specifically, *SEEM* is designed with four desiderata: *i*) Versatility. We introduce a new visual prompt to unify different spatial queries including points, boxes, scribbles and masks, which can further generalize to a different referring image; *ii*) Compositionality. We learn a joint visual-semantic space between text and visual prompts, which facilitates the dynamic composition of two prompt types required for various segmentation tasks; *iii*) Interactivity. We further incorporate learnable memory prompts into the decoder to retain segmentation history through mask-guided cross-attention from decoder to image features; and *iv*) Semantic-awareness. We use a text encoder to encode text queries and mask labels into the same semantic space for open-vocabulary segmentation. We conduct a comprehensive empirical study to validate the effectiveness of *SEEM* across diverse segmentation tasks. Notably, our single *SEEM* model achieves competitive performance across interactive segmentation, generic segmentation, referring segmentation, and video object segmentation on 9 datasets with minimum 1/100 supervision. Furthermore, *SEEM* showcases a remarkable capacity for generalization to novel prompts or their combinations, rendering it a readily universal image segmentation interface.

## 1   Introduction

Image segmentation is arguably the most important yet challenging problem in computer vision. In the past, we have witnessed significant progress in a wide range of segmentation tasks including

37th Conference on Neural Information Processing Systems (NeurIPS 2023).

instance, semantic and panoptic segmentation [1, 2, 3, 4, 5, 6, 7]. Most recently, we are observing a clear trend toward more flexible segmentation models in different aspects: 1) From closed-set to open-vocabulary segmentation. Many recent works proposed to either leverage contrastive learning methods or pretrained multi-modal foundation models (*e.g.*, CLIP [8]) to make the segmentation models more transferable to unseen concepts [9, 10, 11, 12]; 2) From generic to referring segmentation. In addition to generic segmentation that segments an image thoroughly given a predetermined set of concepts, language-based referring segmentation provides a user-friendly way of segmenting a specific region referred by an arbitrary text phrase [13, 14, 15, 16, 17]; and 3) From one-shot to interactive segmentation. In practice, segmentation models do not necessarily produce satisfactory masks in one round. As such, people are also studying how to progressively refine the segmentation results through intimate interactions between humans and models [18, 19, 20, 21].

Despite the aforementioned efforts taken to design more powerful and feasible segmentation models, we are still lacking a universal segmentation interface that is capable of accommodating various types of human prompts and tackling different segmentation tasks as studied in individual works. In contrast, Large Language Models (LLMs) have already emerged as such a universal interaction interface for language tasks, from early models like GPT-3 [22] and T5 [23], to conversational agent [24] augmented by advanced prompting [25, 26, 27] and chain-of-thought [28, 29, 30]. In this work, we strive for a universal interface for *segmenting everything everywhere all at once* in an image. On this interface, we are targeted at unifying all segmentation tasks with a single model in a promptable manner. To achieve this goal, we propose a new prompting scheme in mask decoder that has four important properties: *versatility*, *compositionality*, *interactivity*, and *semantic-awareness*. Specifically, we propose to encode points, masks, text, boxes, and even a referred region from another image into prompts in the same joint visual-semantic space. As such, our model can deal with any combination of the input prompts, leading to strong compositionality. To enable interactivity, we further introduce memory prompts for condensing the previous segmentation information followed by communication with other prompts. As for semantic awareness, our model can provide an open-set semantic label to any output segmentation.

With the proposed prompting scheme, we build a segment-everything-everywhere model called **SEEM** comprised of a simple Transformer encoder-decoder architecture [31, 6] with an extra text encoder [11, 32]. In *SEEM*, the decoding process emulates a generative LLM but with a multimodality-in-multimodality-out interface. An image encoder and text encoder are used as the prompt encoder to encode all types of queries, which are fed into the decoder. Concretely, we encode all spatial queries, namely, points, boxes, scribbles and masks into *visual prompts* by pooling their corresponding visual features from the image encoder, and use the text encoder to convert text queries into *text prompts*. By training on diverse segmentation tasks, our model learns to deal with various prompts, align the visual and text prompts, and promote their synergy via cross-attention between them. As a result, our single model after pretraining attains competitive performance across all segmentation tasks. Since the prompts of all 5 different types are mapped to the *joint visual-semantic space*, we can feasibly combine prompts to resolve the ambiguity to obtain better segmentation results and enable zero-shot adaptation to unseen user prompts. Furthermore, our model can immediately generalize to the case of using an exemplar image segment as the prompt and video object segmentation in a zero-shot fashion. In addition to its strong generalization capability, *SEEM* is also more efficient for interactive segmentation compared with the counterparts like SimpleClick [32]. Since we take the prompts as input to the decoder, when doing multi-round interactions with humans, our model only needs to run the feature extractor once at the beginning and lightweight decoding each per round. To the end, we build a segmentation interface with a single pre-trained model that can segment every object with semantics (everything), cover every pixel in the image (everywhere), and support all possible compositions of prompts (all at once). In summary, our contributions are threefold:

- We design a new prompting scheme that can encode various user intents into prompts in a *joint visual-semantic space*, enabling strong flexibility for various segmentation tasks and generalization capability to unseen prompts or their combinations.

- We build *SEEM*, a universal and interactive segmentation interface that integrates the newly designed prompting mechanism into a lightweight decoder for *all* segmentation tasks, leading to a model possessing properties of versatility, compositionality, interactivity, and semantic awareness.

- We conduct extensive experiments and visualizations to show that our model has strong performance on many segmentation tasks including open-vocabulary generic segmentation, interactive segmentation, referring segmentation, and segmentation tasks with combined prompts.

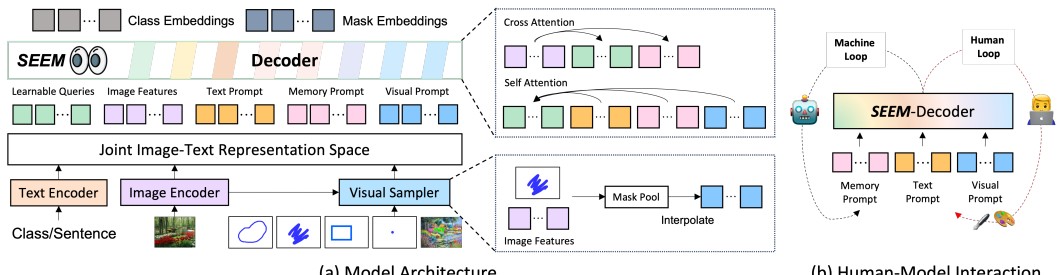

Figure 2: Overview of *SEEM*- Decoder. (a) *SEEM* encodes image, text, and human inputs into *joint visual-semantic space* as queries, features, and prompts, and then decodes queries to class and mask embeddings. (b) With the benefit of *SEEM* decoder, the machine loop enables memorizing history mask information, and the human loop provides new corrections to the next round.

## 2   Related Work

**Interactive segmentation.** Interactive segmentation is the task of segmenting objects by interactively taking user inputs. It has been a longstanding problem and has achieved considerable progress [33, 34, 35, 20, 21, 36]. Generally, the interaction types can take various forms, such as clicks, boxes, polygons, and scribbles, among which click-based interaction models are the most prevalent. Concurrent to our work, SAM [36] proposed a promptable segmentation model trained on 11 million images and 1.1 billion masks. It takes user interactions as prompts for general segmentation. Though SAM demonstrates strong zero-shot performance, it produces segmentations without semantic meaning. In addition, its prompt types are limited to points, boxes, and text, whereas our model can also take in a referred region from another image as a prompt.

**Generic segmentation.** Segmentation of visual concepts has been a persistent challenge in the field of computer vision, as evidenced by its extensive literature [37, 38, 39, 40]. Generic segmentation techniques encompass several subtasks, including instance segmentation, semantic segmentation, and panoptic segmentation [4, 2, 3], each focusing on a different semantic level. For example, semantic segmentation aims to identify and label each pixel within an image based on its corresponding semantic class [41, 6, 42]. On the other hand, instance segmentation involves grouping pixels that belong to the same semantic class into separate object instances [4, 43, 7]. Recently, the Detection Transformer (DETR)[31], a model based on the Transformer [44] architecture, has made significant advances in segmentation [45, 6, 7, 46, 47] tasks. However, these approaches cannot recognize objects absent in the training set, which constrains the model to a limited vocabulary size.

**Unified vision models.** Unified vision models [11, 48, 49, 36, 50] have recently drawn a lot of attention because of their advantage in generalizing to various tasks and flexibility. These models can deal with multiple vision tasks or data distributions. Among them, some [11, 48, 49] train multiple tasks together with only one model and thus can deal with all training tasks without finetuning on each target task. On the other hand, SAM [36] and SegGPT [50] propose training strategies that enable their models to handle new tasks and data distributions in a zero-shot manner. The second approach is more favorable since there is no need to resolve conflicts among tasks during training.

## 3   Method

### 3.1   Model Design

*SEEM* employs a generic encoder-decoder architecture but also employs a sophisticated interaction scheme between queries and prompts, as shown in Fig. 2 (a). Given an input image $\mathbf{I} \in \mathcal{R}^{H \times W \times 3}$, an image encoder is first used to extract image features $\mathbf{Z}$. Then, *SEEM*-Decoder predicts the masks $\mathbf{M}$ and semantic concepts $\mathbf{C}$ based on the query outputs $\mathbf{O}_h^m$ (mask embeddings) and $\mathbf{O}_h^c$ (class embeddings), which interact with text, visual, and memory prompts $\langle \mathbf{P}_t, \mathbf{P}_v, \mathbf{P}_m \rangle$:

$$\langle \mathbf{O}_h^m, \mathbf{O}_h^c \rangle = \mathbf{Decoder}(\mathbf{Q}_h; \langle \mathbf{P}_t, \mathbf{P}_v, \mathbf{P}_m \rangle | \mathbf{Z}) \tag{1}$$

$$\mathbf{M} = \mathbf{MaskPredictor}(\mathbf{O}_h^m) \tag{2}$$

$$\mathbf{C} = \mathbf{ConceptClassifier}(\mathbf{O}_h^c) \tag{3}$$

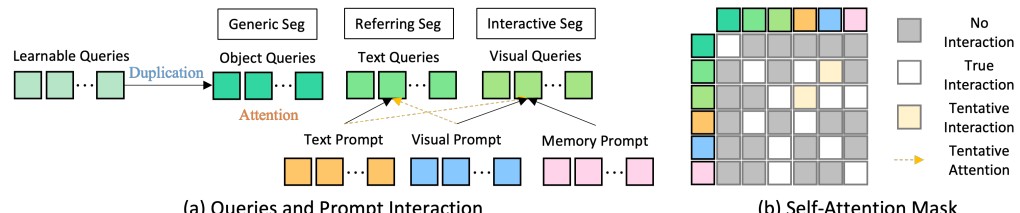

(a) Queries and Prompt Interaction          (b) Self-Attention Mask

Figure 3: Queries and prompt interaction during training and evaluation. (a) Learnable queries are duplicated as object, grounding, and visual queries with the same set of weights for each task. (b) Attention mask between any two kinds of tokens (denoted as $qpm$ in Algorithm. 1). Tentative means the interaction is not trained but able to do inference without any modification.

where $\mathbf{Q}_h$ is the learnable queries, and $\mathbf{P}_t$, $\mathbf{P}_v$, $\mathbf{P}_m$ represent the text prompts, visual prompts, and memory prompts, respectively. During training, $\mathbf{Q}_h$ is duplicated for generic, referring, and interactive segmentation, as shown in Fig. 3. The corresponding prompts interact with their queries through self-attention. The learnable queries can freely interact with all prompts at inference time, thereby enabling zero-shot composition. Our design is inspired by the successful practice in X-Decoder [11]. However, we highlight the differences in Eq. (1), marked in red, which allow for a universal model for image segmentation with the following properties:

**Versatile**. In *SEEM*, we introduce visual prompts $\mathbf{P}_v$ to handle *all* non-textual inputs, such as points, boxes, scribbles, *and* a referred region from another image. These non-textual queries are beneficial to disambiguate the user's intent when textual prompts alone fail to identify the correct segment. For interactive segmentation, previous works either convert spatial queries to masks and feed them into the image backbone [20] or use different prompt encoders for each input type (points, boxes) [36]. The first approach can be too heavy in applications because each interaction requires the image to go through the feature extractor. The second approach is hard to generalize to unseen prompts. To address these limitations, we propose a visual sampler (Fig. 2 (a)) to convert all kinds of non-textual queries to visual prompts that lie in the same visual embedding space:

$$\mathbf{P}_v = \mathbf{VisualSampler}(\mathbf{s}, \hat{\mathbf{Z}}) \tag{4}$$

where $\hat{\mathbf{Z}}$ is the feature maps extracted from either the target image (*i.e.*, $\hat{\mathbf{Z}} = \mathbf{Z}$) or a referred image, and $s \in \{\mathsf{points}, \mathsf{box}, \mathsf{scribbles}, \mathsf{polygons}\}$ are the sampling locations specified by the user. We first pool the corresponding region from the image feature through point sampling [6]. For all visual prompts, we interpolate at most 512 point feature vectors uniformly from the region specified by the prompt. A notable merit of our proposed method is that the visual prompts are naturally well-aligned with the textual prompts, as our model continuously learns a common visual-semantic space through panoptic and referring segmentation.

**Compositional**. In practice, a user may cast their intent using different or combined prompt types. Hence, a compositional approach to prompting is essential for real-world applications. However, we confront two issues during model training. First, the training data usually only covers a single type of interaction (e.g., none, textual, visual). Second, although we use visual prompts to unify all non-textual prompts and align them with textual prompts, their embedding spaces remain inherently different. To mitigate this problem, we propose to match prompts of different types with different outputs. Considering that visual prompts $\mathbf{P}_v$ come from image features while textual prompts $\mathbf{P}_t$ come from the text encoder, we select matched output indices for visual and textual prompts by matching them with the mask embeddings $\mathbf{O}_h^m$ or class embeddings $\mathbf{O}_h^c$, respectively:

$$ID_v \leftarrow \mathbf{Match}(\mathbf{O}_h^m \cdot \mathbf{P}_v + \mathbf{IoU}_{mask}) \tag{5}$$
$$ID_t \leftarrow \mathbf{Match}(\mathbf{O}_h^c \cdot \mathbf{P}_t + \mathbf{IoU}_{mask}) \tag{6}$$

where $\mathbf{IoU}_{mask}$ is the IoU between ground-truth and predicted masks. The proposed separate matching method outperforms approaches that only match with either $\mathbf{O}_h^m$ or $\mathbf{O}_h^c$ for all prompts.

After training, our model becomes familiar with all prompt types and supports a variety of compositions, such as no prompts, one prompt type, or both visual and textual prompts using the same model and weights. *In particular, the visual and textual prompts can be simply concatenated and fed to SEEM-Decoder, even though it was never trained in this way.*

**Interactive**. Interactive segmentation usually cannot be completed in one shot and requires multiple interaction rounds for refinement, similar to conversational agents like ChatGPT. In *SEEM*, we

**Algorithm 1:** Pseudo code for SEEM.

```
# Inputs: Image(img)[B,3,H,W]; Pos_Mask(pm), Neg_Mask(nm)[B,1,H,W]; Text(txt)[abc...];
# Variables: Learnable Queries(Q_h); Attention Masks between Q and P(qpm)
# Functions: Img_Encoder(),Text_Encoder(),Visual_Sampler(),feature_attn(),prompt_attn(),output();

1 def init( ):
2     Q_o,Q_t,Q_v = Q_h.copy();# Initialize object, text and visual queries.
3     F_v,P_t = Img_Encoder(img), Text_Encoder(txt);# F_v and P_t denote image feature, text
          prompt.
4     P_v = Visual_Sampler(F_v, pm, nm);# Sample visual prompt from image feature, pos/neg
          mask.

5 def SEEM_Decoder(F_v,Q_o,Q_t,Q_v,P_v,P_t,P_m):
6     Q_o,Q_t,Q_v = feature_attn(F_v,Q_o,Q_t,Q_v);# Cross attend queries with image features.
7     Q_o,Q_t,Q_v = prompt_attn(qpm,Q_o,Q_t,Q_v,P_v,P_t,P_m);# Self attend queries and prompts.
8     O_m,O_c,P_m = output(F_v,Q_o,Q_t,Q_v);# Compute mask and class outputs.

9 def forward(img,pm,nm,txt):
10    F_v,Q_o,Q_t,Q_v,P_v,P_t = init(); P_m = None;# Initialize variables.
11    for i in range(max_iter):
12        O_m,O_c,P_m = SEEM_Decoder(F_v,Q_o,Q_t,Q_v,P_v,P_t,P_m)
```

propose a new type of prompt called *memory prompts* $\mathbf{P}_m$ and use them to convey the knowledge of the masks from the previous iteration to the current one. Unlike previous works that use a network to encode the previous mask [20, 36], we introduce no extra module but simply a few memory prompts. These memory prompts encode the history information by using a mask-guided cross-attention layer [6]:

$$\mathbf{P}_m^l = \mathbf{MaskedCrossAtt}(\mathbf{P}_m^{l-1}; \mathbf{M}_p | \mathbf{Z}) \tag{7}$$

where $\mathbf{M}_p$ is the previous mask, and $\mathbf{Z}$ is the image feature map. In this way, cross-attention only takes effect inside the regions specified by the previous mask. The updated memory prompts $\mathbf{P}_m^l$ then interact with the other prompts via self-attention to convey the historical information for the current round.

**Semantic-aware**. Different from previous class-agnostic interactive segmentation works such as Simple Click [20] and the concurrent work SAM [36], our model produces semantic labels to masks for all kinds of prompt combinations in a zero-shot manner, since our visual prompt features are aligned with textual features in a *joint visual-semantic space*. As shown in Fig. 3, semantic labels are directly computed using $\mathbf{O}_h^c$ (output of visual queries) and the text embedding. Although **we do not train with any semantic labels for interactive segmentation**, the calculated logits are well-aligned, benefiting from the *joint visual-semantic space*.

### 3.2 Model Pipeline and Loss Functions

We summarize the training and evaluation pipeline of the proposed method with Pytorch-style pseudo-code in Algorithm 1. *SEEM* is trained with a linear combination of losses for panoptic segmentation, referring segmentation, and interactive segmentation:

$$\begin{aligned} \mathcal{L} =& \alpha\mathcal{L}_{\text{c\_CE\_pano}} + \beta\mathcal{L}_{\text{m\_BCE\_pano}} + \gamma\mathcal{L}_{\text{m\_DICE\_pano}} + a\mathcal{L}_{\text{c\_CE\_ref}} + b\mathcal{L}_{\text{m\_BCE\_ref}} \\ &+ c\mathcal{L}_{\text{m\_DICE\_ref}} + a\mathcal{L}_{\text{c\_CE\_iseg}} + b\mathcal{L}_{\text{m\_BCE\_iseg}} + c\mathcal{L}_{\text{m\_DICE\_iseg}} \end{aligned} \tag{8}$$

Where $\alpha = 2, \beta = \gamma = 5, a = 0.2, b = c = 2$, CE, BCE, and DICE denotes cross-entropy, binary cross entropy and dice loss, respectively.

## 4 Experiments

**Datasets and Settings**. *SEEM* is trained on three tasks: panoptic segmentation, referring segmentation, and interactive segmentation. Panoptic and interactive segmentation are trained on COCO2017 [51] with panoptic segmentation annotations. Following [11], we exclude the validation set of Ref-COCOg [52], resulting in 107K segmentation images in total. For referring segmentation, we use a combination of Ref-COCO, Ref-COCOg, and Ref-COCO+ for COCO image annotations. We evaluate generic segmentation (instance/panoptic/semantic), referring segmentation, and interactive segmentation.

**Implementation Details and Evaluation Metrics**. Our model framework follows X-Decoder [11] except the decoder. That is, we have a vision backbone, a language backbone, an encoder, and

Table 1: **One model** for segmentation on a wide range of segmentation tasks. *SEEM* is the first model to simultaneously support generic segmentation, referring segmentation, and interactive segmentation, as well as prompt compositionality. (#Concurrent work. - indicates the model does not have capability for the task, * indicates do not have reported number.)

| Method | Segmentation Data | Type | Generic Segmentation COCO | | | Referring Segmentation RefCOCOg | | | Interactive Segmentation PascalVOC | | | | | |
|---|---|---|---|---|---|---|---|---|---|---|---|---|---|---|
| | | | PQ | mAP | mIoU | cIoU | mIoU | AP50 | 5-NoC85 | 10-NoC85 | 20-NoC85 | 5-NoC90 | 10-NoC90 | 20-NoC90 |
| Mask2Former (T) [6] | COCO (0.12M) | | 53.2 | 43.3 | 63.2 | - | - | - | - | - | - | - | - | - |
| Mask2Former (B) [6] | COCO (0.12M) | | 56.4 | 46.3 | 67.1 | - | - | - | - | - | - | - | - | - |
| Mask2Former (L) [6] | COCO (0.12M) | | 57.8 | 48.6 | 67.4 | - | - | - | - | - | - | - | - | - |
| Pano/SegFormer (B) [45] | COCO (0.12M) | Segmentation | 55.4 | * | * | - | - | - | - | - | - | - | - | - |
| LAVT (B) [53] | Ref-COCO (0.03M) | | - | - | - | 61.2 | * | * | - | - | - | - | - | - |
| PolyFormer (B) [17] | Ref-COCO+VG+... (0.16M) | | - | - | - | 69.3 | * | * | - | - | - | - | - | - |
| PolyFormer (L) [17] | Ref-COCO+VG+... (0.16M) | | - | - | - | 71.1 | * | * | - | - | - | - | - | - |
| RITM (<T) [18] | COCO+LVIS (0.12M) | | - | - | - | - | - | - | * | * | 2.19 | * | * | 2.57 |
| PseudoClick (<T) [54] | COCO (0.12M) | | - | - | - | - | - | - | * | * | 1.94 | * | * | 2.25 |
| FocalClick (T) [21] | COCO (0.12M) | | - | - | - | - | - | - | * | * | 2.97 | * | * | 3.52 |
| FocalClick (B) [21] | COCO (0.12M) | Interactive | - | - | - | - | - | - | * | * | 2.46 | * | * | 2.88 |
| SimpleClick (B) [20] | COCO+LVIS (0.12M) | | - | - | - | - | - | - | 1.75 | 1.93 | 2.06 | 1.94 | 2.19 | 2.38 |
| SimpleClick (L) [20] | COCO+LVIS (0.12M) | | - | - | - | - | - | - | 1.52 | 1.64 | 1.72 | 1.67 | 1.84 | 1.96 |
| SimpleClick (H) [20] | COCO+LVIS (0.12M) | | - | - | - | - | - | - | 1.51 | 1.64 | 1.76 | 1.64 | 1.83 | 1.98 |
| UViM (L) [55] | COCO (0.12M) | | 45.8 | * | * | - | - | - | - | - | - | - | - | - |
| Pix2Seq v2 (B) [56] | COCO (0.12M) | | - | 38.2 | - | - | - | - | - | - | - | - | - | - |
| X-Decoder (T) [11] | COCO (0.12M) | | 52.6 | 41.3 | 62.4 | 59.8 | * | * | - | - | - | - | - | - |
| X-Decoder (B) [11] | COCO (0.12M) | | 56.2 | 45.8 | 66.0 | 64.5 | * | * | - | - | - | - | - | - |
| X-Decoder (L) [11] | COCO (0.12M) | | 56.9 | 46.7 | 67.5 | 64.6 | * | * | - | - | - | - | - | - |
| UNINEXT (T) [48] | Image+Video (3M) | | - | 44.9 | - | 70.0 | * | * | - | - | - | - | - | - |
| UNINEXT (L) [48] | Image+Video (3M) | | - | 49.6 | - | 73.4 | * | * | - | - | - | - | - | - |
| Painter (L) [57] | COCO+ADE+NYUv2 (0.16M) | Generalist | 43.4 | * | * | - | - | - | - | - | - | - | - | - |
| #SegGPT (L) [50] | COCO+ADE+NYUv2 (0.16M) | | 34.4 | * | * | - | - | - | - | - | - | - | - | - |
| #SAM (B) [36] | SAM (11M) | | - | - | - | - | - | - | 2.47 | 2.65 | 3.28 | 2.23 | 3.13 | 4.12 |
| #SAM (L) [36] | SAM (11M) | | - | - | - | - | - | - | 1.85 | 2.15 | 2.60 | 2.01 | 2.46 | 3.12 |
| #SAM (H) [36] | SAM (11M) | | - | - | - | - | - | - | 1.82 | 2.13 | 2.55 | 1.98 | 2.43 | 3.11 |
| SEEM (T) | COCO+LVIS (0.12M) | | 50.8 | 39.7 | 62.2 | 60.9 | 65.7 | 74.8 | 1.72 | 2.30 | 3.37 | 1.97 | 2.83 | 4.41 |
| SEEM (B) | COCO+LVIS (0.12M) | | 56.1 | 46.4 | 66.3 | 65.0 | 69.6 | 78.2 | 1.56 | 2.04 | 2.93 | 1.77 | 2.47 | 3.79 |
| SEEM (L) | COCO+LVIS (0.12M) | | 57.5 | 47.7 | 67.6 | 65.6 | 70.3 | 78.9 | 1.51 | 1.95 | 2.77 | 1.71 | 2.36 | 3.61 |
| SEEM (T) | COCO+LVIS (0.12M) | | - | - | - | 70.4 | 71.7 | 82.1 | 1.72 | 2.28 | 3.32 | 1.97 | 2.82 | 4.37 |
| SEEM (B) | COCO+LVIS (0.12M) | Composition | - | - | - | 76.2 | 77.8 | 87.8 | 1.56 | 2.03 | 2.91 | 1.77 | 2.46 | 3.76 |
| SEEM (L) | COCO+LVIS (0.12M) | | - | - | - | 75.1 | 76.9 | 86.8 | 1.52 | 1.97 | 2.81 | 1.72 | 2.38 | 3.64 |

Table 2: **One model** for all kinds of mask interactions. *SEEM* has strong generalization capability on different input mask types.

| Method | COCO | | | | | Open Image | | | | | ADE | | | | |
|---|---|---|---|---|---|---|---|---|---|---|---|---|---|---|---|
| | Point 1-IoU | Stroke 1-IoU | Scribble 1-IoU | Polygon 1-IoU | Box 1-IoU | Point 1-IoU | Stroke 1-IoU | Scribble 1-IoU | Polygon 1-IoU | BoX 1-IoU | Point 1-IoU | Stroke 1-IoU | Scribble 1-IoU | Polygon 1-IoU | BoX 1-IoU |
| SimpleClick (B) | 49.0 | 33.1 | 65.1 | 48.6 | 42.5 | 48.6 | 29.5 | 54.2 | 49.5 | 42.7 | 47.0 | 19.0 | 52.1 | 48.3 | 37.2 |
| SimpleClick (L) | 38.9 | 33.9 | 68.8 | 39.2 | 34.7 | 37.5 | 29.1 | 59.8 | 35.2 | 31.2 | 36.8 | 16.4 | 56.4 | 41.7 | 29.5 |
| SimpleClick (H) | 59.0 | 37.3 | 71.5 | 45.3 | 52.4 | 54.1 | 32.6 | 62.4 | 39.9 | 41.8 | 52.8 | 18.4 | 58.3 | 46.8 | 41.8 |
| SAM (B) | 58.6 | 22.8 | 34.2 | 44.5 | 50.7 | 62.3 | 28.4 | 39.2 | 45.8 | 53.6 | 51.0 | 21.9 | 31.1 | 31.0 | **58.8** |
| SAM (L) | 64.7 | 44.4 | 57.1 | 60.7 | 50.9 | 65.3 | 45.9 | 55.7 | 57.8 | 52.4 | 57.4 | 45.8 | 53.1 | 45.8 | 58.7 |
| SAM (H) | 65.0 | 27.7 | 30.6 | 37.8 | 50.4 | 27.7 | 26.5 | 29.9 | 41.9 | 52.1 | 58.4 | 20.4 | 22.2 | 28.3 | 58.5 |
| SEEM (T) | 78.9 | 81.0 | 81.2 | 72.2 | 73.7 | 67.1 | **69.4** | **69.5** | 63.1 | **60.9** | 65.4 | 67.3 | 67.3 | 59.0 | 53.4 |
| SEEM (B) | 81.7 | 82.8 | 83.5 | 76.0 | 75.7 | **67.6** | 69.0 | 68.7 | **64.2** | 60.3 | **66.4** | **68.6** | **67.7** | **60.5** | 53.6 |
| SEEM (L) | **83.4** | **84.6** | **84.1** | **76.5** | **76.9** | 66.8 | 67.8 | 67.6 | 62.4 | 60.1 | 65.5 | 66.6 | 66.3 | 58.1 | 54.1 |

*SEEM*-Decoder. For the vision backbone, we use FocalT [58], DaViT-d3 (B), and DaViT-d5 (L) [59]. For the language encoder, we adopt a UniCL or Florence text encoder [60, 61]. For all segmentation tasks, we use standard evaluation metrics: PQ (Panoptic Quality) for panoptic segmentation, AP (Average Precision) for instance segmentation, and mIoU (mean Intersection over Union) for semantic segmentation. For interactive segmentation, we follow previous works [20, 62] to simulate user clicks by comparing the predicted segmentation with the ground-truth one in an automatic way. After one click on the image to generate the predicted mask, the next click is placed at the center of the area with the largest segmentation error. We use the Number of Clicks (NoC) metric to evaluate interactive segmentation performance, which measures the number of clicks needed to achieve a certain Intersection over Union (IoU), i.e., 85% and 90%, denoted as NoC@85 and NoC@90, respectively. We also vary the number of maximum clicks indicated by K-NoC@90 (K=5, 10, 20), and evaluate the mean IoU on the single click denoted as 1-IoU to study the performance on different constraints. More qualitative evaluation with stroke, scribble, polygon, and box as prompts are illustrated in the supplementary material.

## 4.1 Main Results

**Generic segmentation** With one suite of parameters pre-trained on all the segmentation tasks, we are able to evaluate its performance on generic segmentation datasets. As shown in Table 1, *SEEM* maintains competitive panoptic, instance, and semantic segmentation performance against strong baselines. Compared with generalist models such as UViM [55], Pix2Seqv2 [56] and especially the recent model Painter [57] and SegGPT [50], our approach significantly outperforms those methods on generic segmentation with a margin around 10 points on panoptic segmentation metrics.

**Referring segmentation** As shown in Table 1, compared with other referring segmentation and generalist models, *SEEM* achieves competitive performance. Notably, by adding a visual compositional prompt, referring segmentation performance is improved with a large margin by 10.5 cIoU,

Table 3: **Zero-shot** video object segmentation. Without training with video or pairwise image data, our approach is able to do video object segmentation in a zero-shot manner. (#Concurrent work.)

| Method | Segmentation Data | Type | Refer-Type | Zero-Shot | Single Image | DAVIS17 | | | DAVIS16-Interactive | | | YouTube-VOS 2018 | | | | |
|---|---|---|---|---|---|---|---|---|---|---|---|---|---|---|---|---|
| | | | | | | JF | J | F | JF | J | F | G | Js | Fs | Ju | Fu |
| *With Video Data* | | | | | | | | | | | | | | | | |
| AGSS [63] | VOS+DAVIS (0.1M) | | Mask | ✗ | ✗ | 67.4 | 64.9 | 69.9 | - | - | - | 71.3 | 71.3 | 65.5 | 75.2 | 73.1 |
| AGAME [64] | (Synth)VOS+DAVIS (0.11M) | | Mask | ✗ | ✗ | 70.0 | 67.2 | 72.7 | - | - | - | 66.0 | 66.9 | * | 61.2 | * |
| SWEM [65] | Image+VOS+DAVIS (0.25M) | | Mask | ✗ | ✗ | 84.3 | 81.2 | 87.4 | - | - | - | 82.8 | 82.4 | 86.9 | 77.1 | 85.0 |
| XMem [66] | Image+VOS+DAVIS (0.25M) | Video | Mask | ✗ | ✗ | | | | - | - | - | 86.1 | 85.1 | 89.8 | 80.3 | 89.2 |
| SiamMask [67] | COCO+VOS (0.21M) | | Mask | ✗ | ✗ | * | 54.3 | 58.5 | 69.8 | 71.7 | 67.8 | * | 60.2 | 58.2 | 45.1 | 47.7 |
| MiVOS [19] | BL30K+VOS+DAVIS (4.88M) | | Mask/Scribble | ✗ | ✗ | 84.5 | 81.7 | 87.4 | 91.0 | 89.6 | 92.4 | 82.6 | 81.1 | 85.6 | 77.7 | 86.2 |
| ReferFormer-B [68] | RefCOCO(+/g)+VOS+DAVIS (0.13M) | | Text | ✗ | ✗ | 61.1 | 58.1 | 64.1 | - | - | - | * | * | * | * | * |
| TAM-L [69] | XMem+SAM (11.2M) | | Multiple Points | ✗ | ✗ | - | - | - | 88.4 | 87.5 | 89.4 | - | - | - | - | - |
| UNINEXT-T [48] | Image+Video (3M) | Generalist | Mask | ✗ | ✗ | 74.5 | 71.3 | 77.6 | - | - | - | 77.0 | 76.8 | 81.0 | 70.8 | 79.4 |
| UNINEXT-L [48] | Image+Video (3M) | | Mask | ✗ | ✗ | 77.2 | 73.2 | 81.2 | - | - | - | 78.1 | 79.1 | 83.5 | 71.0 | 78.9 |
| UNINEXT-L [48] | Image+Video (3M) | | Text | ✗ | ✗ | 66.7 | 62.3 | 71.1 | - | - | - | * | * | * | * | * |
| *Without Video Data* | | | | | | | | | | | | | | | | |
| Painter-L [57] | COCO+ADE+NYUv2 (0.16M) | | Mask | ✓ | ✗ | 34.6 | 28.5 | 40.8 | - | - | - | 24.1 | 27.6 | 35.8 | 14.3 | 18.7 |
| #SegGPT-L [50] | COCO+ADE+VOC+... (0.25M) | | Mask | ✓ | ✗ | 75.6 | 72.5 | 78.6 | - | - | - | 74.7 | 75.1 | 80.2 | 67.4 | 75.9 |
| #PerSAM-L [70] | SAM+DAVIS (11M) | Generalist | Mask | ✗ | ✓ | 60.3 | 56.6 | 63.9 | - | - | - | * | * | * | * | * |
| SEEM-T | | | | ✓ | ✓ | 60.4 | 57.6 | 63.3 | 62.7 | 58.9 | 66.4 | 51.4 | 55.6 | 44.1 | 59.2 | 46.9 |
| SEEM-B | COCO+LVIS (0.12M) | | Mask/Single Point | ✓ | ✓ | 62.8 | 59.5 | 66.2 | 67.2 | 63.6 | 70.9 | 53.8 | 60.0 | 44.5 | 63.5 | 47.2 |
| SEEM-L | | | | ✓ | ✓ | 58.9 | 55.0 | 62.8 | 62.2 | 58.3 | 66.0 | 50.0 | 57.2 | 38.2 | 61.3 | 43.3 |

Table 4: **Ablation study** on interaction strategy. "#Iter" denotes the maximum training iteration on interactive segmentation in a single forward. "Negative" means adding negative tokens during interactive segmentation. "Scratch" means the model trains from scratch.

| Ablation | Fix | #Iter | Pos | Neg | COCO | | | Referring Segmentation | | | Pascal VOC | | DAVIS17 | | |
|---|---|---|---|---|---|---|---|---|---|---|---|---|---|---|---|
| | | | | | PQ | mAP | mIoU | cIoU | mIoU | AP@50 | NoC50 | NoC90 | JF | J | F |
| Baseline | Y | 0 | ✓ | ✗ | 50.7 | 39.5 | 60.8 | 57.9 | 63.3 | 71.6 | 1.74 | 5.43 | 59.6 | 55.8 | 63.5 |
| - LVIS | ✓ | 2 | ✓ | ✓ | 51.0 | 39.8 | 62.2 | 58.6 | 63.9 | 72.6 | 1.57 | 4.91 | 59.5 | 55.9 | 63.1 |
| + Negative | ✓ | 0 | ✓ | ✓ | 50.9 | 39.8 | 61.4 | 58.8 | 64.0 | 72.6 | 1.81 | 5.41 | 60.1 | 56.3 | 63.9 |
| + Scratch | ✗ | 3 | ✓ | ✓ | 50.2 | 39.5 | 60.7 | 51.4 | 59.2 | 67.0 | 1.45 | 4.41 | 60.6 | 57.7 | 63.4 |
| + Iter | ✓ | 1 | ✓ | ✓ | 50.7 | 39.7 | 60.5 | 58.3 | 63.4 | 71.3 | 1.76 | 5.14 | 59.2 | 55.4 | 63.0 |
| | ✓ | 2 | ✓ | ✓ | 50.5 | 39.5 | 61.0 | 58.0 | 63.2 | 71.6 | 1.78 | 5.20 | 59.6 | 56.2 | 63.0 |
| | ✓ | 3 | ✓ | ✓ | 50.4 | 39.5 | 61.0 | 58.0 | 63.0 | 71.5 | 1.55 | 4.67 | 59.9 | 56.4 | 63.5 |
| | ✓ | 5 | ✓ | ✓ | 50.6 | 39.4 | 60.9 | 58.4 | 63.4 | 71.6 | 1.54 | 4.59 | 59.7 | 56.3 | 63.1 |

6.0 mIoU, and 9.3 AP50 points for the tiny model. And this gap is retained for the base and large model. Specifically, this number is computed by class embeddings $O_h^c$ (Output-Q-Textual). The margin is even larger when computed with mask embeddings $O_h^m$ (Output-Q-Visual) as shown in Table 5. Further, we benchmark the vanilla composition (Ensemble) that directly combines visual and text mask output probabilities as shown in Table 5 row 2.

**Interactive segmentation** As shown in Table 1, our approach achieves comparable performance with the specialized models, e.g. RITM, SimpleClick, and better performance than SAM [36] (B) which is trained with $\times 100$ more segmentation data than ours. Notably, unlike existing interactive models, *SEEM* is the first interface that supports not only classical segmentation tasks but also a wide range of user input types, including text, points, scribbles, boxes, and images, providing strong compositional capabilities as shown in Table 2,5.

Table 5: The term 'Text/Visual Prompt' refers to the modality of information utilized in the study. 'Output Query' is indicative of the type of query employed to predict the output. 'Composition Approach' specifies the method through which text and visual information are integrated.

| Text Prompt | Visual Prompt | Output Query | Composition Approach | Focal-Tiny | | | Davit-Base | | | Davit-Large | | |
|---|---|---|---|---|---|---|---|---|---|---|---|---|
| | | | | cIoU | mIoU | AP@50 | cIoU | mIoU | AP@50 | cIoU | mIoU | AP@50 |
| Y | N | Text | N/A | 58.4 | 63.4 | 71.6 | 63.0 | 68.2 | 76.7 | 62.4 | 67.6 | 75.3 |
| Y | Y | All | Ensemble | 63.0 | 60.0 | 66.9 | 69.3 | 66.6 | 74.3 | 68.9 | 65.5 | 72.7 |
| Y | Y | Text | Self-Attn | 66.5 | 69.6 | 78.8 | 75.0 | 76.9 | 86.3 | 73.2 | 76.5 | 85.9 |
| N | Y | Visual | N/A | 70.7 | 71.8 | 81.3 | 75.4 | 77.8 | 87.4 | **75.2** | 78.2 | 87.7 |
| Y | Y | Visual | Self-Attn | **71.5** | **72.8** | **82.2** | **75.9** | **78.3** | **87.7** | 74.9 | **78.4** | 87.7 |

**User input type of interactive segmentation** In Table 2, we compare 1-IoU of *SEEM* with other strong baselines SimpleClick and SAM with 5 common types of prompts on three datasets. 1-IoU indicates the mean IoU of all images with a single click. The prompt types include point, stroke, scribble, and box. The results show that our *SEEM* achieves the best performance in the extremely limited number of clicks over all three datasets.

**Video object segmentation** Without any modification, our model is able to do (interactive) video object segmentation in a zero-shot manner through the visual prompt (by replacing the current image visuals prompt with the visual prompts from another image). As shown in Table 3, without any observation of DAVIS/VOS dataset [71, 72], our approach is able to achieve close performance in a zero-shot manner with a fully supervised method on DAVIS17 dataset [72]. Meanwhile, our model is able to do interactive video object segmentation on DAVIS16-Interactive [72] and achieves comparable performance with the supervised baselines with one single click of the first frame.

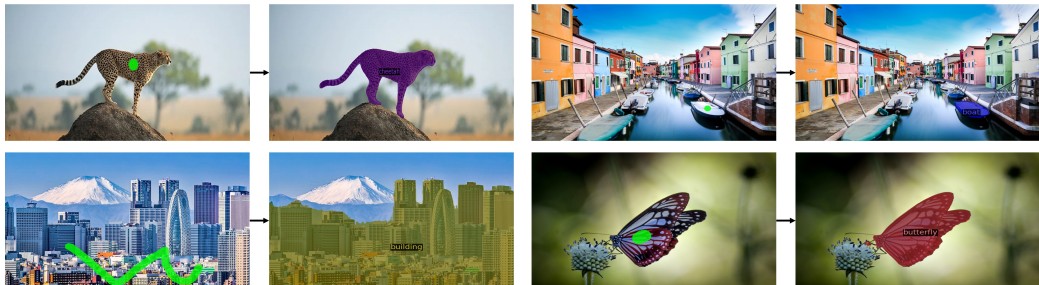

Figure 4: Click/scribble-based segmentation. *SEEM* supports arbitrary formats of clicks or scribbles by users. Moreover, it simultaneously gives the semantic label for the segmented mask, which is not possible in SAM [36].

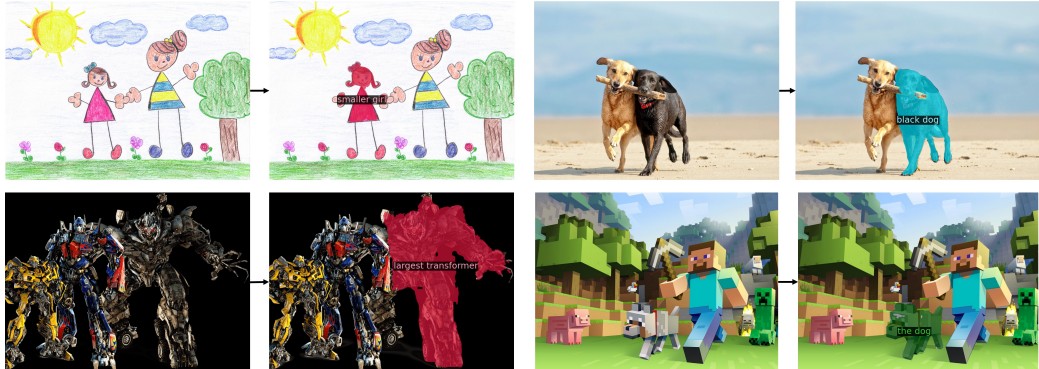

Figure 5: Text to mask or text referring segmentation. The referred text is shown on the masks. *SEEM* adapts to various types of input images in the domain of cartoons, movies, and games.

## 4.2 Ablation Study

We conduct an ablation study on all the training segmentation tasks and zero-shot video object segmentation, dissecting each component of our model. The results are presented in Table 4.

*LVIS mask annotation will improve interactive segmentation results.* We replace the COCO mask with an overlap IoU larger than 0.7 with LVIS mask during training. This will improve the performance on interactive segmentation with 0.3 and 0.2 point gain on NoC0.9 and NoC0.85.

*Training from scratch only hurts referring segmentation performance.* We compare the *SEEM* model trained with X-Decoder pre-trained checkpoint or the checkpoint initialized with UniCL or Florence vision and language backbone (+Scratch). It indicates that training from scratch will slightly improve the performance on interactive segmentation but hurt the referring segmentation performance.

*Increase interactive training iterations does help.* As shown in Table 4, increasing the training iteration (the first N-1 iteration is without gradient) from 1 to 5 will gradually improve the interactive segmentation performance from 5.41 to 4.59 on NoC0.9. As the computation cost increases with more clicks, we use iteration 3 for the main paper results.

## 4.3 Qualitative Results

We further qualitatively evaluate *SEEM*. Based on the proposed prompting scheme and decoder design, with the same suite of parameters, *SEEM* supports a wide range of visual input types.

**Visual prompt interactive segmentation**. In Fig. 4, we show the visualization of using *SEEM* to segment objects in an interactive way. The user can segment objects of interest by simply clicking or drawing a scribble. Taking these prompts, *SEEM* can simultaneously produce both masks and semantic labels for the objects. Note that our model is open-vocabulary, which empowers it to label unseen categories when given the candidate vocabulary (i.e., cheetah and butterfly in Fig. 4). When no vocabulary is given, *SEEM* can segment in a class-agnostic manner.

**Text referring segmentation**. We show the text referring to segmentation visualization results in Fig. 5. The results demonstrate that our model is semantic-aware of open-vocabulary concepts and

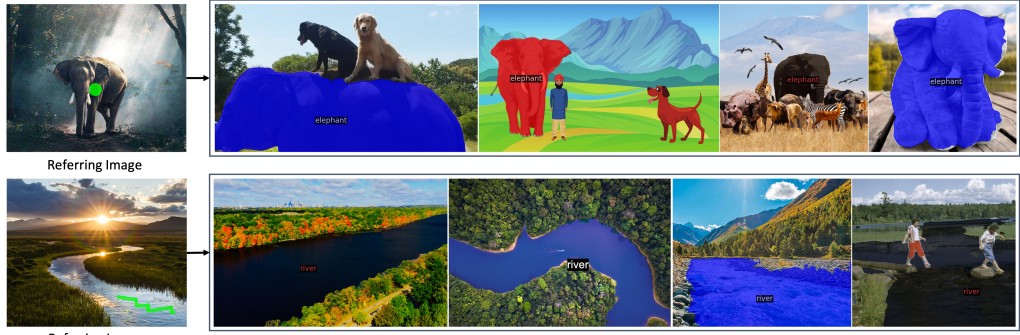

Figure 6: **Zero-shot** visual referring segmentation with *SEEM*. Given a referring image with simple spatial hints, *SEEM* can segment the regions which are semantically similar in different target images.

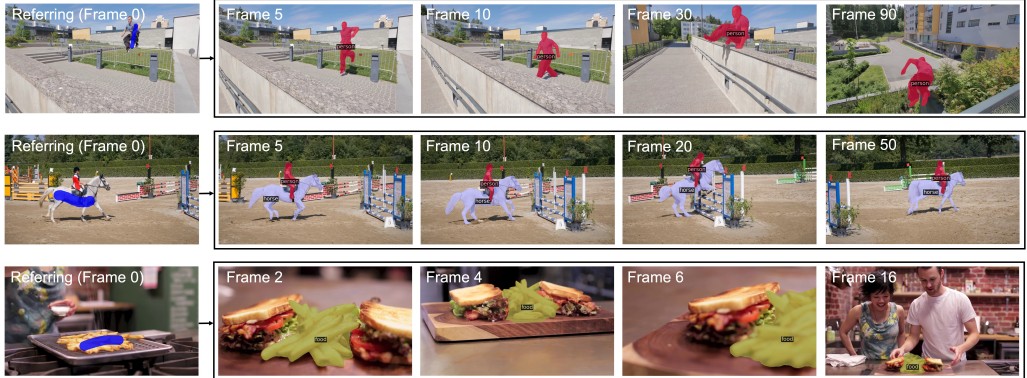

Figure 7: **Zero-shot** video object segmentation using the first frame plus one stroke. From top to bottom, the videos are "parkour" and "horsejump-low" from DAVIS [73], and video 101 from YouCook2 [74]. *SEEM* precisely segments referred objects even with significant appearance changes caused by blurring or intensive deformations.

attributes to understand language. In addition, *SEEM* is able to generalize to unseen scenarios like cartoons, movies, and games.

**Visual referring segmentation**. In Fig.6, we show *SEEM*'s segmentation results when prompted with referring regions from another image. By simply drawing a click or scribble on one referring image, *SEEM* can take it as input and segment objects with similar semantics on other images. Notably, this referring segmentation has a powerful generalization capability to images of other domains. For example, by referring to the elephant in the forest, another object of the same category can be segmented well under drastically different scenes like cartoons, plush toys, and grassland.

**Video object segmentation**. In Fig. 7, we further show *SEEM*'s referring segmentation ability on the video object segmentation task in a zero-shot manner. By referring to the objects in the first frame with scribbles, *SEEM* can precisely segment the corresponding objects in the following frames, even when the following objects change in appearance by blurring or intensive deformations.

## 5 Conclusion

We presented *SEEM*, which can segment everything (all semantics) everywhere (all pixels) all at once (all possible prompt compositions). Apart from performing generic open-vocabulary segmentation, *SEEM* can interactively take different types of visual prompts from the user, including click, box, polygon, scribble, text, and referring region from another image. These visual prompts are mapped into a *joint visual-semantic space* with a prompt encoder, which makes our model versatile to various prompts and can flexibly compose different prompts. Extensive experiments indicate that our model yields competitive performance on several open-vocabulary and interactive segmentation benchmarks. Further studies revealed the robust generalization ability of our model in accurately segmenting images based on diverse user intents. We hope our work will serve as a stepping stone toward a universal and interactive interface for image segmentation and beyond.

**Acknowledgements.** We would like to express our gratitude to Lei Zhang for his generous support. And express the appreciated for the valuable suggestions from Zhenyuan Yang, and discussion with Xiaoyu Xiang. In addition, this work was supported in part by NSF CAREER IIS2150012, NASA 80NSSC21K0295, the Institute of Information and communications Technology Planning and Evaluation (IITP) grant funded by the Korea government (MSIT) (No. 2022-0-00871, Development of AI Autonomy and Knowledge Enhancement for AI Agent Collaboration).

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
