# Supplementary: Segment Everything Everywhere All at Once

## A    Clarification of Main Paper

In main.Table.1, we only marked SegGPT as concurrent work. However, **#SAM is also a concurrent work to ours.**

## B    Architecture details

**Human-Model Interaction** For the Human-Model Interaction depicted in main paper Fig.2.b, users have the flexibility to provide various inputs. These inputs can include positive/negative points, arbitrary shape strokes of the input image, or a referring image. These inputs are encoded as a visual prompt using the visual sampler. Additionally, a user can also input a description text for an object, which will be encoded through a text encoder. The visual and text prompts can be integrated to generate the final output. If the predicted mask does not meet the user's expectations, the user can input corrections in any format, including both visual and text prompts. The previous mask is stored in a memory prompt. The visual prompt can indicate either a correct or incorrect region, marked as positive and negative respectively described in detail in the section below.

**Visual & Text Prompt Interaction** Taking inspiration from [1, 2], our SEEM decoder also employs a hierarchical structure. Our positive and negative prompts are pooled using a visual sampler, which utilizes features at different scales. These features align with image features through a cross-attention mechanism. This design choice helps bridge the gap in the embedding domain at each decoder layer, ensuring that the query, image, and prompt features are synchronized in the Joint Image Text Representation space. Detailed operation of the visual sampler is provided in the pseudo code below.

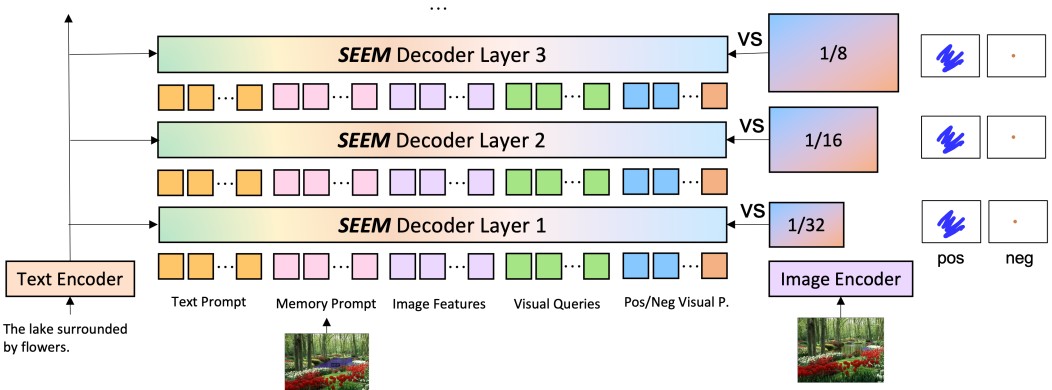

Figure 1: Detailed depiction of the Visual & Text Prompt Interaction in the SEEM Decoder. (VS denotes visual sampler – details are provided in Algorithm 1 in this document.)

**Algorithm 1** Pseudo code for visual sampler in SEEM.

```
# Inputs: img_f - image feature,[B,C,H,W]; pm/nm - pos/neg mask, [B,1,H,W]; max_len=512
# Functions:  import torch.nn.functional as F; import torch
1 def Visual_Sampler(img_f, pm, nm, max_len):
2     pos_emb = img_f[pm]; neg_emb = img_f[nm];# Pool pos/neg features from image feature.
3     pos_emb = F.interpolate(pos_emb, min(max_len, len(pos_emb)), "nearest");# random selection.
4     neg_emb = F.interpolate(neg_emb, min(max_len, len(neg_emb)), "nearest");# random selection.
5     visual_prompt = torch.cat(pos_emb,neg_emb);# Formulate Visual Prompt.
6     visual_queries = torch.mean(pos_emb, dim=0, keepdims=True);# Formulate Queries Spatial.
```

## C Implementation details

### C.1 Training Details

**Dataset Preparation.** For training, we employ the COCO dataset. Guided by the methodologies in [3, 4, 5], we replace overlapping masks in COCO and LVIS that have an Intersection over Union (IoU) value exceeding 0.7 with the corresponding LVIS mask. To emulate user inputs, we make use of the OpenCV library to create scribbles, strokes, polygons, and points. The shape and thickness of these elements are randomly adjusted during training, as is the number of masks. This process constitutes the mask initialization, with sample masks presented in Figure 2. Upon mask initialization, user input is simulated by utilizing the center points of false negative and false positive mask regions. Throughout training, the first N-1 iterations of interactive segmentation are performed without gradient updates. Here, the output mask serves as the input mask for the final layer, simulating human input, where N is a randomly selected number between 0 and 5.

| Scribble | Stroke | Polygon | Point |
|---|---|---|---|

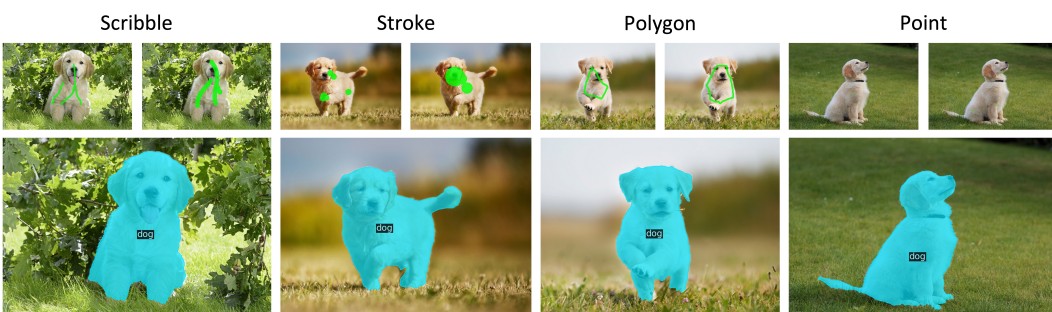

Figure 2: Visualization of mask initialization during training.

**Hyper-parameters.** In our study, we primarily follow the training hyperparameters set in [2]. We conducted training under two settings in the main paper: one that utilizes a vision language pretrained model starting from scratch, and another that employs an X-Decoder pretrained model. When training from a vision language model, we train all parameters with the exception of the language encoder. Conversely, when training from the X-Decoder checkpoint, we only train the SEEM-Decoder. The training duration differs between versions; for the tiny model, the first version requires approximately three days of training on 32 V100 GPUs, whereas the second version demands around one day and five hours on the same GPU configuration.

### C.2 Evaluation Datasets for Interactive Segmentation

The evaluation datasets for interactive segmentation are selected with the primary principle of following existing works whenever possible. When this isn't feasible, we randomly sample 600 images and corresponding masks from the dataset, ensuring a diverse vocabulary distribution. For the COCO dataset, we align with [6] to download the COCO mini dataset comprising around 600 images. For the PASCAL VOC dataset, we adhere to [3, 7] and employ the original PASCAL VOC dataset. Similarly, for the DAVIS dataset, we procure the video frames as chosen in [6]. With regards to all input types assessed in Table 2 of the main paper, we follow the mask generation procedure illustrated in Figure 2.

## C.3 Evaluation Metrics

For interactive segmentation, we use two metrics NOC and 1-IoU:

**NOC:** Number of Clicks (NOC) is commonly used in interactive segmentation. It calculate the average number of clicks required to output a high-quality mask which has an IoU above a predefined threshold. In some cases, a model cannot output masks that exceed the IoU threshold leading to infinite or very large NOC. Therefore, we also cap the largest number of clicks. For example 5-NOC85 denotes the NOC to achieve IoU=85 and the max possible NOC is 5.

**1-IoU:** In most cases, our model outputs high-quality masks in one round of interaction. Therefore, we use 1-IoU to denote the IoU between the output mask and the GT mask with a single prompt.

For video object segmentation (VOS) we mainly report JF [8] metric which is the average of J (Region Similarity) and F (Contour Accuracy).

**Region Similarity J:** In order to quantify the similarity of region-based segmentation, specifically the extent of mislabeled pixels, we utilize the Jaccard index J. This index is defined as the intersection-over-union between the estimated segmentation and the ground-truth mask. The Jaccard index, introduced in PASCAL VOC2008 [9], has gained widespread acceptance due to its ability to provide intuitive and scale-invariant information about the number of mislabeled pixels. For a given output segmentation M and its corresponding ground-truth mask G, the Jaccard index is calculated as $J = \frac{|M \cup G|}{|M \cap G|}$.

**Contour Accuracy F:** From a contour-based perspective, the set of closed contours c(M) can be seen as defining the spatial boundaries of the mask. To assess contour accuracy, one can compute precision and recall ($P_c$ and $R_c$) between the contour points of c(M) and c(G) using a robust bipartite graph matching approach. The F-measure (F) is considered a balanced metric between the two, and is defined as $F = \frac{2P_c R_c}{P_c + R_c}$. In our experiments, we adopt an efficient approximation of the bipartite matching using morphology operators.

# D  Experiments

## D.1  Quantitative Results

**Ablation Study on Modality Composition** We rectify Main.Table.5 by utilizing our latest model that covers all backbone scales, from tiny to base and large. Please note that in the main paper's Table.5, we inadvertently used an older configuration for the Tiny model that deviates from the configurations used in all other evaluations in the main paper. The results depicted in the table below elucidate several key points: (a) Merely ensembling the predicted visual and text masks does not generally enhance the performance of the referring segmentation. (b) Utilizing self-attention to amalgamate visual and text information (refer to Tentative Attention in Main.Figure.3) significantly improves performance across all backbone scale sizes in comparison to using solely text information, regardless of the output query we employ. (c) Generally speaking, supplementing text information with spatial data confers a larger improvement than the converse scenario, suggesting that spatial information tends to yield more accurate results than text information.

Table 1: The term 'Text/Visual Prompt' refers to the modality of information utilized in the study. 'Output Query' is indicative of the type of query employed to predict the output. 'Composition Approach' specifies the method through which text and visual information are integrated.

| Text Prompt | Visual Prompt | Output Query | Composition Approach | Focal-Tiny | | | Davit-Base | | | Davit-Large | | |
|---|---|---|---|---|---|---|---|---|---|---|---|---|
| | | | | cIoU | mIoU | AP@50 | cIoU | mIoU | AP@50 | cIoU | mIoU | AP@50 |
| Y | N | Text | N/A | 58.4 | 63.4 | 71.6 | 63.0 | 68.2 | 76.7 | 62.4 | 67.6 | 75.3 |
| Y | Y | All | Ensemble | 63.0 | 60.0 | 66.9 | 69.3 | 66.6 | 74.3 | 68.9 | 65.5 | 72.7 |
| Y | Y | Text | Self-Attn | 66.5 | 69.6 | 78.8 | 75.0 | 76.9 | 86.3 | 73.2 | 76.5 | 85.9 |
| N | Y | Visual | N/A | 70.7 | 71.8 | 81.3 | 75.4 | 77.8 | 87.4 | **75.2** | 78.2 | 87.7 |
| Y | Y | Visual | Self-Attn | **71.5** | **72.8** | **82.2** | **75.9** | **78.3** | **87.7** | 74.9 | **78.4** | **87.7** |

**Ablation Study on Backbone Architecture.** To facilitate a fair comparison of the model backbone with SAM [10], SegGPT [11], and SimpleClick [4], we employ the SAM pretrained ViT backbone and present quantitative results in Table 2. Several conclusions can be drawn from these observations: (1) The utilization of the SAM pretrained ViT backbone tends to enhance the performance of interactive segmentation, while it may decrease the performance in the domains of generic and referring segmentation. This can be potentially attributed to the fact that the vision backbone is not trained

92 with any semantic labels. (2) As noted in the table, our ViT-L is trained with fix vision backbone
93 because of memory limitation. Compare with ViT-B generic segmentation result is extremely weak.
94 However surprisingly, the referring segmentation result is still comparable or even better.

Table 2: Ablation study on backbone architecture. '*COCO+LVIS' denotes that the model does not strictly adhere to this configuration, given that the SAM pretrained checkpoint is trained with the SAM dataset. We were unable to train the model until completion, thus we have only compared results for the same epoch. Also, '*ViT-L' denotes the backbone is fixed due to limited GPU memory. Full results will be posted in a future version.

| Backbone | Segmentation Data | Epoch | COCO | | | Ref-COCOg | | | VOC | | |
|---|---|---|---|---|---|---|---|---|---|---|---|
| | | | PQ | mAP | mIoU | cIoU | mIoU | AP@50 | NoC@50 | NoC@85 | NoC@90 |
| davit-d3 | COCO+LVIS | 27/50 | 50.6 | 41.6 | 61.4 | 49.2 | 57.9 | 64.7 | 1.54 | 3.45 | 4.43 |
| ViT-B | SAM+COCO | 27/50 | 48.5 | 40.3 | 56.8 | 50.1 | 59.0 | 66.1 | 1.49 | 2.99 | 3.91 |
| davit-d5 | COCO+LVIS | 27/50 | 54.4 | 44.4 | 64.7 | 51.0 | 58.8 | 65.4 | 1.46 | 3.21 | 4.03 |
| *ViT-L | SAM+COCO | 27/50 | 44.4 | 38.0 | 52.3 | 51.5 | 59.8 | 66.6 | 1.42 | 2.91 | 3.71 |

## D.2 Qualitative results

95

96 **Open Vocabulary Interactive Segmentation.** In Main.Figure.4, we evaluated interactive segmenta-
97 tion within a closed vocabulary, identical to the COCO classes. In Figure 3, we examine this process
98 within an open vocabulary setting, manually providing candidate classes within the given image.
99 Remarkably, our model demonstrates adept performance on unseen classes. This capability is a
100 legacy of the X-Decoder and is powered by the Joint Image-Text Representation Space in SEEM.
101 Notably, certain items such as yellow corn, leaf, french fries, orange juice, and others, which have
102 never been trained, are successfully recognized by our model.

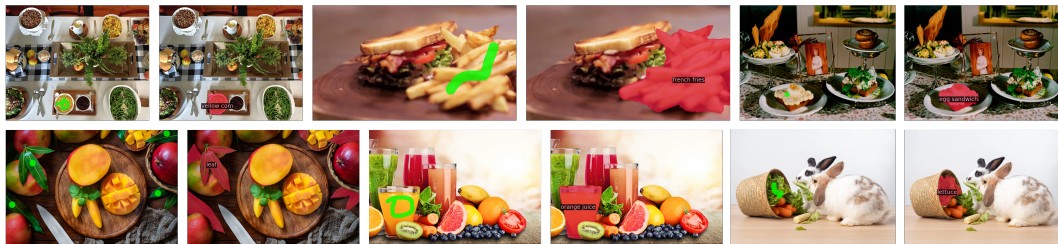

Figure 3: Visualization on Open Vocabulary Interactive Segmentation. (Best view with zoom in)

103 **Multi-Round Interactive Segmentation.** In the main paper, as seen in Figure.4, we evaluated the
104 single-shot result with an arbitrary input stroke provided by a human. However, in Figure 4, we assess
105 interactive segmentation in a multi-round format, illustrating the correction procedure of SEEM.
106 Images are grouped in sets of four following this sequence: Input Round1, Output1, Input Round2,
107 and Output2. Positive strokes are colored in green, while negative strokes are colored in blue. The
108 results clearly demonstrate that our negative tokens can effectively remove undesirable masks from
109 the previous round, and positive tokens are capable of inpainting missing parts from the previous
110 round.

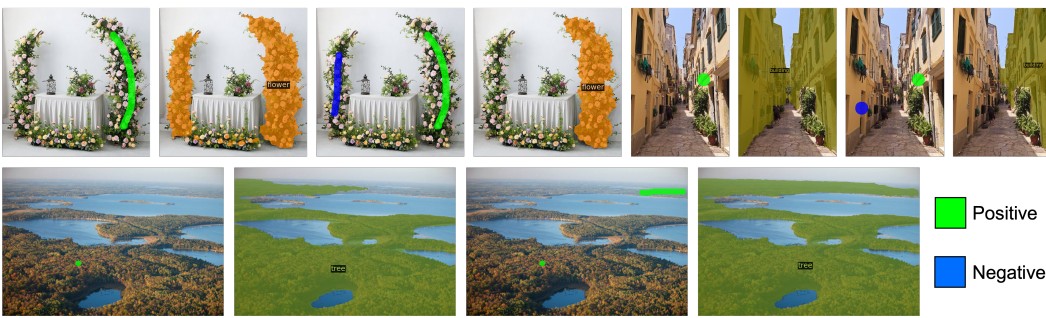

Figure 4: Multi-Round Interactive Segmentation.

**Modality Composition.** As demonstrated in the main paper, specifically in Figures 4 and 5, our model is capable of performing both interactive and referring segmentation. We have also shown in the main paper's Table 5, and in Table 1 of the supplementary material, that the composition of spatial and textual information can enhance performance quantitatively. As illustrated in Figure 5, we present visualization results that integrate spatial and textual content. Our spatial content efficiently resolves ambiguities inherent in the text. For example, it helps determine which dog is playing with the yellow ball, or identify what the left blue magical creature is.

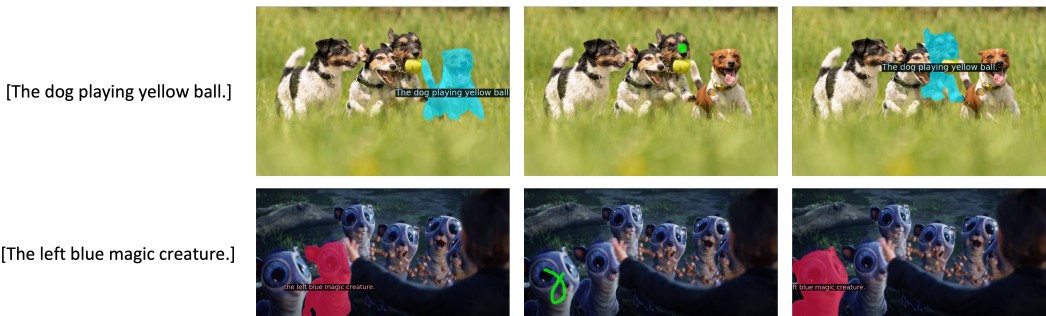

Figure 5: Modality Composition. (best viewed with zoom in.)

**Comparison of Interactive Segmentation with Concurrent Work.** While we have showcased numerous capabilities of the SEEM model independently, here we examine how it fares against concurrent works on interactive segmentation. We contrast our work with both SAM and SegGPT. As evident in the column of SAM output, the model seems to struggle with either identifying the specific object (e.g., which cat the user is pointing to) or capturing all components of an object (e.g., missing legs of a chair). Additionally, as SegGPT is trained with referring images, it appears that the model may falter when partial masks are provided. In summary, our model holds the advantage in identifying objects, and it also exhibits a proficiency in classifying instances, an aspect that seems to be absent in the other concurrent methods.

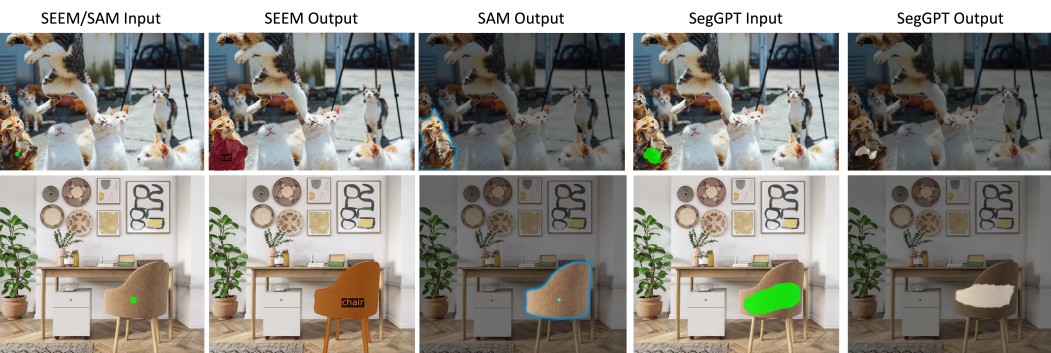

Figure 6: Comparison of Interactive Segmentation with concurrent work.