# OpenReview forum: "Segment Everything Everywhere All at Once"
_NeurIPS.cc/2023/Conference — NeurIPS 2023 poster_

### Official Review · Reviewer_gMmy · 2023-07-01

**Soundness:** 3 good
**Presentation:** 3 good
**Contribution:** 3 good
**Rating:** 6
**Confidence:** 5

**Summary:**

The paper proposes SEEM, a method that unifies several segmentation tasks. The network takes text and/or different kinds of visual prompts as inputs. The model then outputs the corresponding segmentation masks for the referred objects. In particular, SEEM trains a unified prompt embedding space. A set of learned queries then alternately attends to image features and the prompt embeddings. Finally, the output embeddings are matched to the prompts and decoded into masks.

**Strengths:**

-	Unifying several segmentation tasks.
-	Impressive results on most tasks.
-	Similar or better than SAM in many cases, despite less training data.
-	Reasonably well written.

**Weaknesses:**

1.	Some parts of the method lack motivation and details. In particular, the attention mask used in the prompt attention is not well motivated. It is also not well explained that (and how) the output embeddings are matched to the prompts.

2.	The paper is heavily derived from X decoder. The magnitude of the contributions is not clear. The authors do not discuss the relation to X decoder. In fact it is hardly mentioned in the paper, yet the method is completely based on it. Seems like the main new addition here is the visual prompt encoder.

3.	The results for VOS are quite poor. This indicates that the method is not as robust to object appearance variations, which is crucial in many tasks.

4.	The authors should report training and inference time and memory requirements.

I am leaning positive for the paper. However, the authors need to address my concerns above.

**Questions:**

See weaknesses.

Moreover:
Is the method robust to the case when the prompted text does not match any object in the image?

**Limitations:**

Limitations are not discussed in the main paper, but should be mentioned.

---

> ### Author Rebuttal · Authors · 2023-08-10
>
> **Q1: Lack motivation and details (e.g. self-attention mask, output embeddings).**
>
> Sorry for the confusion, we explain the motivation of self-attention mask in details here.
>
> * Inputs: Text, user input interactive scribbles, and none.
> * SEEM Decoder Inputs: (1) Queries. (2) Prompts. (3) Image Features.
> * SEEM Decoder Operation: (1) Cross-Attention between Queries and Image Features. (2) Self-Attention between Queries and prompts. (3) Compute outputs using Queries.
>
> The self-attention mask shown in Fig.3 (b) are modeling the interaction between queries and prompts. As we are computing output masks and semantics using queries, *the self-attention between queries and prompts link between inputs and queries, so that the queries understand where to attend during cross-attention with image features to output the final results.* And this is the prompt attention defined in L7 of Algorithm 1.
>
> Then, we illustrate the details in how to compute the output embeddings. Given the queries that have attended with image features and prompts, we project it separately using mask and class embedding layers (L112-123). The final mask and semantic class are computed by the similarity with 1/4 scale image features and language embeddings.
>
> **Q2: Novelty in comparison with X-Decoder.**
> Good question. We appreciate the reviewers point out this very related work that we are not discussed sufficiently. In comparison with X-Decoder, SEEM has the following difference:
>
> * Technique
>
>     (1) The model design is different. As marked with red font in Equation.1, SEEM has visual and memory prompts other than text prompts. Visual prompts can have various formats such as clicks, scribbles, boxes and referred images. Memory prompts are used to memorize history interactions and support multi-round interactive segmentation.
>
>     (2) The training method is different. Different from X-decoder which train on all the masks of an image in an iteration, SEEM trains on one mask for multiple rounds in one iteration to enable multi-round interaction.
>
>     (3) Human Interactive Animation. We construct an interaction segmentation dataset to animate human interactions such as scribbles and clicks as shown in Appendix C1.
>
> * Capability:
>
>     (1) Modality Composition. Our model is able to combine prompts of different modalities for prompting segmentation due to the joint representation space for text and visual prompts.
>
>     (2) Emerging property. Without training, our model shows the capability of image referred segmentation and VOS. Surprisingly, SEEM performs well even when source and target images for referring are in different styles.
>
> **Q3: Performance on VOS in comparison with other methods.**
>
> Thanks for the interesting question, VOS indeed is a very important application. However, the main contribution of SEEM is lied on interactive segmentation, and zero-shot prompt composition, referring image segmentation. While VOS is an emerging property.
>
> As a zero-shot method, although we are not able to out-perform the previous approach that is fully supervised on both dataset and task. However, we achieve better and comparable performance with Painter-L and SegGPT-L that are trained on paired image data. Even without supervised cross-image referencing, SEEM is able to out-perform Painter-L with a large margin (53.8 vs. 24.1 on G metrics in YT-VOS).
>
> Comparing SEEM with SegGPT is an apple to pear comparison, for example in coco panoptic segmentation, we are able to out perform Painter and SegGPT with a large margin.
>
> |               | Pretrained    | Segmentation Data        | PQ   | mAP  | mIoU |
> |---------------|---------------|--------------------------|------|------|------|
> | Painter (L)   | *             | COCO+ADE+NYUv2 (0.16M)   | 43.4 | -    | -    |
> | SegGPT (L)    | Painter (L)   | COCO+ADE+VOC+... (0.25M) | 34.4 | -    | -    |
> | X-Decoder (L) | *             | COCO                     | 56.9 | 46.7 | 67.5 |
> | SEEM (L)      | X-Decoder (L) | COCO+LVIS (0.12M)        | 57.5 | 47.7 | 67.6 |
>
> In addition, when the input prompt is not complete enough to conver the whole object, our tracking performance is much better than SegGPT, which indicates that our model can take more types of prompt and better generalization ability. We also show the visualization results in the rebuttal PDF.
>
> **Q4: Training and evaluation time and memory cost.**
>
> |          | Resolution | #Param | FPS/img | Training Memory | Inference Memory |
> |----------|------------|--------|---------|-----------------|------------------|
> | SEEM (T) | 1024       | 101.1M | 9.0     |                 | 3124M            |
> | SEEM (B) | 1024       | 140.7M | 6.6     |                 | 3184M            |
> | SEEM (L) | 1024       | 415.3M | 3.7     |                 | 4612M            |
>
> We compute the number on 1 V100 gpu with a single image with resolution 1024 x 1024.
>
> **Q5: Generalization capability on text not included in the image.**
>
> Yes, our method is robust to the case when the prompted text does not match any object in the image. Our matching is implemented by computing the similarities between the prompt embedding and object features. We are able to use a threshold to filter those object features with very low similarities.

---

> > ### Author Response · Authors · 2023-08-10
> > **Clarification of Q4 and Q5**
> >
> > **Q4: Training and evaluation time and memory cost.**
> >
> > |          | Resolution | #Param | FPS/img | Training Memory | Inference Memory |
> > |----------|------------|--------|---------|-----------------|------------------|
> > | SEEM (T) | 1024       | 101.1M | 9.0     | 4804M           | 3124M            |
> > | SEEM (B) | 1024       | 140.7M | 6.6     | 5556M           | 3184M            |
> > | SEEM (L) | 1024       | 415.3M | 3.7     | 6588M           | 4612M            |
> >
> > We compute the FPS on 1 V100 gpu with a single image with resolution 1024 x 1024. Both the training memory and inference memory are referring to batch size equal to 1. Specifically for model training, we fix the visual encoder and language encoder, only the SEEM-Decoder part is learnable.
> >
> > **Q5: Generalization capability on text not included in the image.**
> >
> > Yes, our method is robust to the case when the prompted text does not match any object in the image. Our matching is implemented by computing the similarities between the prompt embedding and object features. We are able to use a threshold to filter those object features with very low similarities. The task we are evaluating is referring segmentation.
> >
> > To further demonstrate this aspect quantitatively, we run our model SEEM-Large on COCO val 2017 of 5000 images. For each of the 5000 images, we sample a positive text prompt and a negative text prompt, and perform the text-based referring segmentation. For each prompt, we calculate the scores over $K$ object tokens using the following calculations:
> >
> > $v = v / norm(v) \in \mathcal{R}^{K \times d}; t = t / norm(t) \in \mathcal{R}^{1 \times d}$
> > $score = t \cdot v^t / scale \in \mathcal{R}^{1 \times K}$
> > $score_{max} = \max(score, dim=1)$
> >
> > where $v$ is the $K$ object query embeddings, $t$ is the textual embedding, and $d$ the feature dimension. We use the maximal score $score_{max}$ as the matching score.
> >
> > Based on the above calculations, we obtain 5000 matching scores for positive prompts and negative ones, respectively. Then we compute the mean and standard deviation as shown below:
> >
> > | Text Prompt Type | Mean | Std |
> > | -------- | -------- | -------- |
> > | Positive     | 17.88     | 1.43     |
> > | Negative     | 7.36     | 3.00     |
> >
> > Clearly, there is a huge margin between the matching scores for positive and negative text prompts, which justify our above statement.

---

> > > ### Comment · Reviewer_gMmy · 2023-08-13
> > >
> > > I thank the authors for their responses.
> > >
> > > **Q1:** I understood these things after going back and forth over the manuscript a few times. My point was that, I think the authors could explain these things in a better and more clear way in the paper. This would make the method more easy to comprehend.
> > >
> > > **Q2:** It is quite clear that the authors directly builds on top of the X decoder. Also if looking at the similarity of presentation. This is not itself an issue. The issue is that it seems that the authors want to hide this. I think this is also harms the readability of the paper. In fact, the proposed method by the authors made much more sense to me after I head the X decoder paper in detail. I therefore would strongly suggest the authors to add a "Preliminaries" section or similar, first briefly describing the X decoder idea. I think this would greatly benefit the presentation of the method, as well as the motivation of the contributions.
> > >
> > > I think that the contributions wrt X decoder is sufficient.
> > >
> > > **Q5:** I thank the authors for this experiment. However, it would be more interesting if the authors could provide an analysis of the percentage of frames where a negative prompt lead to a detected mask (for some good global value of the threshold). Or to provide some precision-recall analysis.
> > >
> > > In addition to text prompt, it would also be interesting if the authors could perform a similar evaluation for image prompts (VOS or visual referring segmentation).

---

> > > > ### Author Response · Authors · 2023-08-14
> > > >
> > > > Thanks for the prompt reply : ) Below are some quick responses, we are working on the experiment and will update shortly for Q5.
> > > >
> > > > **Q1**: Thanks for your efforts on going through the paper iteratively, and this is a very valuable suggestion. We will definitely include a more clear explanation on the attention mask and output embeddings in the future versions. To make sure our work is reproducible for the community, we will release the training/evaluation code and models.
> > > >
> > > > **Q2**: We sincerely apologize for making the paper seem to be presented in this way and thank you for recognizing the contribution of our work with respect to X-Decoder. It is a great advice to add a "preliminary" section about X-Decoder before presenting our SEEM. We will follow the suggestion and carefully integrate this into the next version. In addition, we would like to clarify that it is an honest mistake instead of we are pretending not building SEEM upon X-Decoder. Again we appreciate that you point this out and give us a chance to have a better presentation for future versions.
> > > >
> > > > **Q5**: Great suggestion! we will provide the results shortly.

---

> > > > > ### Author Response · Authors · 2023-08-15
> > > > >
> > > > > **Q5**: We are providing the accuracy of the threshold that is able to successfully separate the positive and negative example in cross-validation format on both text prompt and visual prompt.
> > > > >
> > > > > 5-Fold cross validation results for text referring segmentation:
> > > > >
> > > > > Validation Fold | Fold-1 | Fold-2 | Fold-3 | Fold-4 | Fold-5 |
> > > > > | -------- | -------- | -------- | -------- |  -------- |  -------- |
> > > > > EER Threshold | 13.57     | 13.77     | 13.74     | 13.67 | 13.67 |
> > > > > Accuracy | 98.43 | 98.84 | 98.94 | 98.23 | 98.54 |
> > > > >
> > > > > 5-Fold cross validation results for image referring segmentation:
> > > > > Validation Fold | Fold-1 | Fold-2 | Fold-3 | Fold-4 | Fold-5 |
> > > > > | -------- | -------- | -------- | -------- |  -------- |  -------- |
> > > > > EER Threshold | -0.14     | -0.19     | -0.19     | -0.19 | -0.19 |
> > > > > Accuracy | 74.17 | 70.83 | 72.50 | 73.33 | 71.67 |
> > > > >
> > > > > * We are using DAVIS17 for evaluating the image referring segmentation, the positive object are sampled from the same video of the same object id, and the negative object are randomly sampled from another video.
> > > > >
> > > > > EER - Equal Error Rate
> > > > >
> > > > > Accuracy - The accuracy of positive and negative examples given the EER Threshold
> > > > >
> > > > > The result clearly shows that our model is able to separate the positive and negative example given a threshold.

---

> > > > > > ### Author Response · Authors · 2023-08-20
> > > > > >
> > > > > > Hi Reviewer gMmy01,
> > > > > >
> > > > > > As suggested, we have provided solution for all the questions you raised including the newly required experiments on "the percentage of frames where a negative prompt led to a detected mask."
> > > > > >
> > > > > > Since the due for the final recommendation is close, it would be really appreciated that **you could confirm whether the questions are resolved and make your final recommendation**. Thanks!

---

> > > > > > > ### Comment · Reviewer_gMmy · 2023-08-22
> > > > > > >
> > > > > > > I thank the authors for the responses. I am happy with the response and results and will increase my rating. I think that the authors should integrate the above feedback and experiments to the main paper and/or supplementary material.

---

> > > > > > > > ### Author Response · Authors · 2023-08-22
> > > > > > > >
> > > > > > > > Thanks so much! We will make the suggested adjustment on the incoming updates!

---

### Official Review · Reviewer_x17t · 2023-07-06

**Soundness:** 4 excellent
**Presentation:** 4 excellent
**Contribution:** 3 good
**Rating:** 8
**Confidence:** 4

**Summary:**

This paper introduces SEEM, an innovative and interactive model for segmentation. SEEM stands out with its versatility in handling various prompts, such as points, boxes, scribbles, masks, and texts. The model's design is elegantly simple yet highly effective. Additionally, SEEM incorporates memory prompts to preserve segmentation history, enhancing its interactivity and overall performance.

**Strengths:**

1. The paper is well-presented, showcasing clear and concise writing. The authors effectively communicate their ideas.

2. The proposed idea is both innovative and highly effective. The designed model, despite its simplicity, showcases remarkable capabilities when dealing with a wide range of prompts. Its versatility and adaptability make it be applicable with more tasks.

3. The experimental results are solid. Through a combination of qualitative and quantitative analyses, the authors demonstrate the effectiveness of their idea.

**Weaknesses:**

1. Do the authors observe whether training on all types of prompts performing better than on one type or a subset types of prompts? or which types of prompt performs best and which one still lacks of improvement.  It would be better to perform this ablation because it can give more insights to the society.

2. since this is a very general framework supporting many different types of prompts, and it can be more easily extended with more training data.  I would also like to see its scalability with larger amount of supervised training data if there is sufficient computing resources.

**Questions:**

Do the authors plan to train on larger amount of dataset? (such as the one trained on SAM or other rich datasets with the semantic labels).  It would be very interesting to see how SEEM scale up with large amount of supervised or weakly supervised labels.

**Limitations:**

not mentioned

---

> ### Author Rebuttal · Authors · 2023-08-09
>
> **Q1: Ablation study on prompt type.**
>
> Thanks for the valuable suggestion, ablation the prompt type is intuitive to study the effectiveness of each counter part, here are the results:
>
> |          | Panoptic | Grounding | Interactive |             | COCO |      |      | Ref-COCOg |      |      | VOC      |          |
> |----------|----------|-----------|-------------|-------------|------|------|------|-----------|------|------|----------|----------|
> |          | Vanilla  | Text      | Visual      | Multi-Round | PQ   | mAP  | mIoU | mIoU      | cIoU | AP50 | 20-NoC50 | 20-NoC90 |
> | SEEM (T) | Y        | Y         | Y           | X           | 51.2 | 39.8 | 61.8 | 63.8      | 58.7 | 72.6 | 1.82     | 5.51     |
> | SEEM (T) | Y        | Y         | X           | X           | 51.0 | 39.9 | 61.1 | 64.2      | 58.8 | 72.4 | -        | -        |
> | SEEM (T) | Y        | X         | X           | X           | 50.8 | 39.9 | 60.7 | -         | -    | -    | -        | -        |
>
> We gradually remove the visual prompt and text prompt from SEEM decoder, and here we only trained on a single round on Interactive Segmentation for fast training purpose. From the table, we can clearly observe that while gradually add addition capabilities to the model, both text and visual prompt will improve panoptic segmentation. And add interactive segmentation will not influence referring segmentation task.
>
> **Q2: Scaling up training data.**
>
> Thank you for your insightful suggestion, scaling up SEEM is for a good direction. Although with the short-time frame we are not able to fully train on large scale dataset such as Object365, or SA-1B. However, we are able to take advantage of the pretrained checkpoint on SA-1B. By using the SA-1B pretrained backbone, we are able out-perform our paper reported number with a good margin as shown below:
>
> |          | Backbone | Multi-Queries in Single Inference | 20-NoC@50 | 20-NoC@85 | 20-NoC90 | 1-IoU |
> |----------|----------|------------------------------------|-----------|-----------|----------|-------|
> | SAM (B)  | SAM-B    | X                                  | /         | 3.28      | 4.12     | /     |
> | SAM (L)  | SAM-L    | X                                  | /         | 2.60      | 3.12     | /     |
> | SAM (H)  | SAM-H    | X                                  | /         | 2.55      | 3.11     | /     |
> | SEEM (B) | Davit-d3 | X                                  | /         | 2.93      | 3.79     | /     |
> | SEEM (L) | Davit-d5 | X                                  | /         | 2.77      | 3.61     | /     |
> | SEEM (B) | Davit-d3 | Y                                  | 1.37      | 2.71      | 3.43     | 87.2  |
> | SEEM (L) | Davit-d5 | Y                                  | 1.38      | 2.70      | 3.47     | 88.5  |
> | SEEM (B) | SAM-B    | Y                                  | 1.29      | 2.53      | 3.21     | 86.2  |
> | SEEM (L) | SAM-L    | Y                                  | 1.30      | 2.42      | 2.97     | 84.5  |
>
> Here, Multi-Queries in Single Inference means that we are able to forward multiple interactive query in a single forward. This new capability will not add any training or inference overhead but will improve inference efficiency and accuracy. Our approach now safely surpasses SAM with a good margin, even compare with their huge model.
>
> In addition, as acknowledged by the reviewer, our model holds the potential for scaling up data due to its compatibility with a diverse range of prompts. Some large-scale datasets that can be harnessed for training encompass SA-1B (consisting of 11 million images, each annotated with masks) and Objects365 (comprising 1.7 million images, annotated with semantic information). To effectively leverage Objects365, which exclusively offers bounding box annotations for training, we can employ SAM to generate mask annotations based on the provided bounding box in Objects365. It is important to note that accommodating such voluminous data for training demands an extended period, coupled with the intricacies of harmonizing diverse data formats inherent in disparate datasets. Therefore, we leave this to future work.

---

> ### Author Response · Authors · 2023-08-14
>
> Hi reviewer x17t, thanks for your reviews again! And hope the rebuttal material solve your confusion.
>
> It would be really appreciated if you are willing to give any additional comment, and feel free to drop any question if you have any confusion on the rebuttal material : )

---

> > ### Comment · Reviewer_x17t · 2023-08-14
> >
> > Thanks for your effort during the rebuttal.
> >
> > The experimental results are very satisfying and promising.  I will slightly improve my rating after the rebuttal.
> > If the authors have time, It would be better to scale up the the model with larger dataset and share the results with the whole community, although it is not necessary to do it during the rebuttal due to the limited time.

---

> > > ### Author Response · Authors · 2023-08-15
> > >
> > > Thanks for your encouraging comment! We will try our best scale up SEEM to larger dataset, it is both beneficial to the community and our work itself : )

---

### Official Review · Reviewer_R6GZ · 2023-07-07

**Soundness:** 3 good
**Presentation:** 3 good
**Contribution:** 3 good
**Rating:** 5
**Confidence:** 5

**Summary:**

[Task] In this work, authors introduc SEEM, a promptable and interactive model designed for comprehensive image segmentation. SEEM aims to segment all objects in an image simultaneously, addressing various segmentation tasks. The key contribution of SEEM is its novel decoding mechanism, which allows for diverse prompting and behavior similar to large language models (LLMs).

[Method] SEEM is built with four key objectives: versatility, compositionality, interactivity, and semantic-awareness. The model incorporates a visual prompt that unifies different spatial queries such as points, boxes, scribbles, and masks, enabling generalization to different referring images. It also learns a joint visual-semantic space for dynamic prompt composition in different segmentation tasks. Additionally, SEEM incorporate learnable memory prompts in the decoder to retain segmentation history and utilizes mask-guided cross-attention from the decoder to image features. Moreover, a text encoder is employed to encode text queries and mask labels into the same semantic space, facilitating open-vocabulary segmentation.

[Experiments] Empirical studies validate the effectiveness of SEEM across diverse segmentation tasks. The model achieves competitive performance in interactive segmentation, semantic segmentation, referring segmentation, and video object segmentation across nine datasets, requiring only minimal 1/100 supervision. SEEM demonstrates remarkable generalization capability to novel prompts or their combinations, positioning it as a versatile and universal image segmentation interface.

**Strengths:**

[highlight the trend towards more flexible segmentation models] In this work, the authors address the need for a universal segmentation interface capable of handling different types of human prompts and addressing various segmentation tasks. They highlight the trend towards more flexible segmentation models, including open-vocabulary segmentation, referring segmentation, and interactive segmentation. Inspired by the success of Large Language Models (LLMs) as universal interaction interfaces for language tasks, the authors propose SEEM, a promptable and interactive model for segmenting everything everywhere all at once in an image.

[technical contribution] The proposed approach, SEEM, introduces a novel prompting scheme in the mask decoder with four key properties: versatility, compositionality, interactivity, and semantic-awareness. By encoding different types of prompts (points, masks, text, boxes, and referred regions) into a joint visual-semantic space, SEEM achieves strong compositionality and can handle any combination of input prompts. Memory prompts are introduced to retain previous segmentation information and enable interactivity. The model also provides open-set semantic labels for output segmentation.

[tasks and results] The model is trained on diverse segmentation tasks to learn how to handle different prompts, align visual and text prompts, and promote their synergy through cross-attention. The single pre-trained SEEM model achieves competitive performance across all segmentation tasks, leveraging the joint visual-semantic space for prompt combination and zero-shot adaptation. In addition to its strong generalization capability, SEEM is efficient for interactive segmentation compared to other methods. Overall, SEEM offers a segmentation interface with a single pre-trained model that can handle all types of prompts, segment every object with semantics, and cover every pixel in the image.

**Weaknesses:**

[Model performance] The model performance of SEEM on certain benchmarks, including ADE20K for open-vocabulary panoptic segmentation and DAVIS for video instance segmentation, is often lower, and in some cases, significantly lower than the baselines X-Decoder, ODISE, and UNINEXT. It is worth noting that SEEM's framework follows a similar approach to X-Decoder.
While the concept of a universal and interactive segmentation interface is intriguing, it appears that the model performance on open-vocabulary benchmarks is compromised in SEEM. It would be beneficial for the authors to provide an explanation for this discrepancy.

[Unfair comparisons] In Table 2, there are some comparisons that seem unfair for two reasons. Firstly, the evaluation of SEEM on COCO, which provides instance-level annotations, aligns better with SEEM's training pipeline that includes instance-level annotations. However, this evaluation method puts SAM at a disadvantage since it is trained to respect hierarchical segmentation, including whole-part-subpart relationships. Secondly, SEEM is already trained on in-domain data from COCO's training set, while SAM is pretrained on out-of-domain data. This discrepancy in training data can impact the performance comparison between the two models. To ensure fair comparisons, it would be beneficial to evaluate both models using evaluation protocols that align with their respective training pipelines and take into account the domain of the training data.

**Questions:**

[Handling dataset imbalance] Since SEEM is trained on multiple datasets, there might be an imbalance in the number of samples per dataset. I'm curious to know if adjusting the frequency of training samples from each dataset has any impact on the model's performance. Can the authors shed some light on whether they have explored techniques such as data augmentation or sample weighting to address this issue? Additionally, it would be interesting to understand if adjusting the training sample frequency has any effect on mitigating the impact of dataset imbalance and improving the overall model performance.

**Limitations:**

Yes, the authors adequately addressed the limitations

---

> ### Author Rebuttal · Authors · 2023-08-08
>
> **Q1: Performance on open-vocabulary segmentation is lower in comparison with X-Decoder, and performance on video instance segmentation performance on DAVIS is lower than UNINEXT.**
>
> We agree with the reviewer that our open-vocabulary performance on ADE20k is *potentially* lower than X-Decoder and ODISE, and the VIS performance is lower than UNINEXT. While all these methods are toward a unified architecture, the underlying goal and training data are very different.
>
> |           | Input Modality                  | Output Modality   | Composition      | Emerging Property                                         | Segmentation Data  | Ensemble |
> |-----------|---------------------------------|-------------------|------------------|-----------------------------------------------------------|--------------------|----------|
> | SEEM      | Vision & Language & Human Input | Vision            | Task & Embedding | Referring Image Segmentation, Video Instance Segmentation | COCO (0.12M)       | N/A      |
> | X-Decoder | Vision & Language               | Vision & Language | Task             | N/A                                                       | COCO (0.12M)       | N/A      |
> | ODISE     | Vision                          | Vision            | N/A              | N/A                                                       | COCO (0.12M)       | Y        |
> | UNINEXT   | Vision & Language & Video       | Vision            | N/A              | N/A                                                       | Image + Video (3M) | N/A      |
>
> (1) Open vocabulary Segmentation: In comparison to X-Decoder, ODISE, we agree with the reviewer that we are potentially lower. However, X-Decoder are trained on 4M more image text pairs, while ODISE have heavily fixed ensemble modules (Implicit Captionor, Mask Generator, Diffusion UNet), while only the alignment is tuned. The SEEM objective is never trained towards open-vocabulary segmentation than X-Decoder and ODISE.
>
> One interesting thing is that we never provide any quantitative open-vocabulary evaluation in neither main paper nor supplementary material. To resolve the confusion, here we provide the full evaluation table. we can clearly observe that although SEEM never trained towards open-vocabulary segmentation objective, however, it retains resonable open-vocabulary capability with X-Decoder pretrained checkpoint. In addition, for the model initialized with SAM ViT-B backbone that doesn't have any vision-language pretraing than UniCL, it still perform comparable with X-Decoder-Seg.
>
> |                    | COCO  |      |         | Others           | ADE20K |      |      |
> |--------------------|-------|------|---------|------------------|--------|------|------|
> |                    | Class | Mask | Caption | Image-Text Pairs | PQ     | mAP  | mIoU |
> | x-Decoder-Seg (B)  | Y     | Y    | X       | X                | 15.3   | 8.3  | 19.5 |
> | x-Decoder-Seg+ (B) | Y     | Y    | Y       | X                | 16.9   | 9.5  | 23.8 |
> | X-Decoder (B)      | Y     | Y    | Y       | Y                | 21.1   | 11.7 | 27.2 |
> | SEEM (davit-B)     | Y     | Y    | X       | X                | 16.7   | 11.6 | 23.5 |
> | SEEM (samvit-B)    | Y     | Y    | X       | X                | 13.3   | 9.5  | 18.9 |
>
> (2) Video Object Segmentation: **UNINEXT is explicitly trained with video data and even on VOS dataset**, where our approach is only trained on single image and coco dataset (4th column in Table.3).  For the models that does not trained on Video dataset but with pair images (Painter-L), SEEM nearly doubled the performance (34.6 vs. 62.8 on JF Davis17, 24.1 vs 53.8 on G VOS18). Noted that we never trained on pairwise images. And even compared with the model tuned on in-domain dataset (PerSAM-L with DAVIS data), SEEM could also surpass the baseline with a margin (62.8 vs. 60.3 JF on DAVIS). These results all showed that our approach has strong zero-shot and task generalization capability.
>
> **Q2: Unfair Comparison with SAM on in-domain dataset COCO**
>
> In both Table.1 and Table.2, we compare SEEM and SAM on VOC, OpenImage and ADE20k dataset (these are out-of-domain datasets) in addition to COCO on both interactive segmentation and first click IoU (1-IoU). COCO is only one of four dataset that we are evaluating, and our model is trained on COCO only and never see other datasets like OpenImage, VOC and etc..
>
> **Q3: Training dataset imbalance and joint training protocol**
>
> The problem on joint dataset training is very interesting. However, as shown in Table. 1 Main Paper (L188-189) we never trained on multiple datasets in SEEM, we only trained on COCO dataset that is 1/100 data of SAM.
>
> For the discussion of joint dataset training itself, X-Decoder and UNINEXT are trained on multiple datasets. In X-Decoder, it trained on COCO and 4M image text pairs while it takes 32 COCO images and 1024 image text pairs in one training iteration. That strategy is chosen by task, where COCO are trained 50 epoch in Mask2Former, and image-text pair loss usually need 1024 batch size to avoid performance drop. In addition to sampling frequency, balancing loss on different tasks also plays an important role on joint training. In addition to train the model in a single batch one pass, UNINEXT are taken use of both object level and pixel level image data, while also leverage video data. In addition to balance the training data in a single batch, they also train different data in different round that means their model is trained on object detection data in the first round, segmentation data in the second round, while for video segmentation on the third round.

---

> > ### Comment · Reviewer_R6GZ · 2023-08-15
> >
> > Thank you for your comprehensive response addressing the questions raised during the rebuttal period. I would also like to extend my gratitude to the authors for clarifying some of the confusion I had regarding this paper. Having reviewed the rebuttal, I find that the majority of my questions have been effectively resolved. Consequently, *I am inclined to recommend the acceptance of this paper.*
> >
> > While I acknowledge the significant interest inherent in many of the findings presented in this work, I still have reservations regarding the performance gap between X-Decoder and SEEM, particularly given that SEEM is built upon X-Decoder. Echoing questions raised by other reviewers, I propose that training the model on a large-scale dataset and reporting the results would be nice. I am intrigued to learn whether increased data for training could yield further improvements in model performance.

---

> > > ### Author Response · Authors · 2023-08-15
> > >
> > > Hi Reviewer R6GZ, thanks so much for your encouraging comments! We agree with your comment that while building upon X-Decoder, SEEM has its own advantages (composition capability, emerging property and etc. ) and disadvantages (overall open vocabulary segmentation performance). However, we emphasis that the disadvantages are coming from the task focusing. We have done some pilot study on scaling up segmentation dataset, that has shown generalization ability of our approach (improved NoC, 1-IoU). Thanks again for the reviewer's reply : )

---

> > > ### Author Response · Authors · 2023-08-20
> > >
> > > Hi reviewer R6GZ,
> > >
> > > Thanks so much for your review and positive comments!
> > >
> > > We appreciated that you are considering "inclined to recommend the acceptance of this paper" during the discussion period. The current rating is still "Borderline accept", it would be really appreciated if you could **make the final recommendation before the deadline on Aug 21st 1 pm EDT** : )
> > >
> > > Thanks for your patience again!

---

> ### Author Response · Authors · 2023-08-14
>
> Hi reviewer R6GZ, thanks for your reviews again! And hope the rebuttal material solve your confusion.
>
> It would be really appreciated if you are willing to give any additional comment, and feel free to drop any question if you have any confusion on the rebuttal material : )

---

### Official Review · Reviewer_cQYZ · 2023-07-29

**Soundness:** 3 good
**Presentation:** 3 good
**Contribution:** 3 good
**Rating:** 7
**Confidence:** 4

**Summary:**

This paper proposes a universal segmentation model, SEEM, for all segmentation tasks. A visual sampler module unifies different kinds of human inputs and they are encoded into a joint visual-semantic space together with image and text so that the SEEM model can learn semantic labels for masks. The proposed SEEM model achieves competitive performance across interactive segmentation, generic segmentation, referring segmentation, and video object segmentation. Besides, SEEM shows a strong ability to generalization to novel prompts or their combinations.

**Strengths:**

1. The proposed model has superior performance and has achieved competitive results across different types of segmentation tasks.
2. The proposed model can accept a variety of visual prompts input types and has a strong generalization ability, which makes it possible to be applied to more scenarios.
3. Different from the previous segmentation model, the proposed model can learn the semantic labels corresponding to the segmentation masks, thus enabling open vocabulary segmentation.

**Weaknesses:**

1. In this paper, when comparing the performance of the proposed model with the previous models, only the scales of training data are compared, without the comparison of parameters and calculation. This may cause some doubts about the comparison fairness.
2. The analysis of some results is insufficient. In the ablation study section, the performance changes caused by the change of variables are not consistent in different segmentation tasks. Only the changing trend of the results is given without analyzing the potential reasons.

**Questions:**

1. Although SEEM can learn the semantic labels of masks, it is limited to the categories of COCO. Is there any way to expand the scope of semantic labels without introducing a large amount of extra data?
2. Can you provide a comparison with other universal segmentation models in terms of the number of model parameters and calculations? If this comparison is not applicable, what are the reasons?

**Limitations:**

Limitations are not explicitly discussed in this paper. The scale of training data is limited, and it does not support partial segmentation as in SAM (Segmenting Anything Model).

---

> ### Author Rebuttal · Authors · 2023-08-02
>
> **Q1: Insufficient Analysis for ablation study.**
>
> Thanks for the suggestion. We have analyzed the ablation study in Section 4.2 of the main paper. To make it more comprehensive, we further summarize and analyze the main findings in Table 4 below:
>
> *Removing LVIS mask annotations (Row 2 vs. Row 6)* from the training data will improve generic segmentation, and referring segmentation performance, but decrease the interactive segmentation performance. That is caused by the fact that although LVIS segmentation mask is more accurate than coco (improves performance on interactive segmentation), however, its mask has a domain gap with coco annotation (decrease on generic and referring segmentation).
>
> *Training from scratch (Row 4 vs. Row 7)* that (1) UniCL pretrained weight instead of X-Decoder pretrained weight is used, (2) the vision backbone and transformer encoder are unfixed. It will not influence generic segmentation performance, but decrease the referring segmentation performance, and improve interactive segmentation, vos performance. That is caused by the fact that X-Decoder weight actually provides better region-language alignment than UniCL pretrained weights. Thus the performance drop on referring segmentation is reasonable. In addition, as the weight of the vision backbone and transformer encoder weight is tuned, it will give better performance on interactive segmentation tasks that is newly proposed in SEEM as more weight can be adapted to the new task.
>
> *Increasing the training iteration (Row 5-8)* on interactive segmentation will not influence the performance on generic segmentation, referring segmentation and vos, while improving the interactive segmentation result consistently. That is caused by the fact that the first n-1 iterations will only forward interactive segmentation without any gradient. Also adding training iterations that mimic the human interaction with images will improve performance when evaluating multi-round IoU.
>
> We will include the above detailed analysis in our revision.
>
> **Q2: Generalize to open-vocabulary interactive segmentation.**
>
> We are able to generalize to more semantic labels without further training as shown in the example images in Fig.3 of supplementary material (e.g. Yellow Corn, Leaf, Orange Juice, etc. are never trained in coco dataset). This behavior is acquired by two reasons:
>
> (1) We leveraged language encoders that are pretrained with language-image contrastive loss (CLIP or UniCL), which renders more generalizable language embeddings for a wide range of visual concepts.
>
> (2) We used the pre-trained X-Decoder as the initialization and only fine-tuned the decoder part so that it retains open-vocabulary capability.
>
> In addition to the qualitative results, here we evaluate on ADE20K dataset and compared it with prior work X-Decoder. From the table, we can clearly observe that although SEEM never trained towards an open-vocabulary segmentation objective, however, it retains reasonable open-vocabulary capability with X-Decoder pretrained checkpoint. In addition, for the model initialized with SAM that doesn't have any vision-language pretraining such as UniCL, it still performs comparably with X-Decoder-Seg that uses the same amount of annotations.
>
> |                    | Backbone  | COCO  |      |         | Others           | ADE20K |      |      |
> |--------------------|-----------|-------|------|---------|------------------|--------|------|------|
> |                    | Pretrain  | Class | Mask | Caption | Image-Text Pairs | PQ     | mAP  | mIoU |
> | x-Decoder-Seg (B)  | UniCL     | Y     | Y    | X       | X                | 15.3   | 8.3  | 19.5 |
> | x-Decoder-Seg+ (B) | UniCL     | Y     | Y    | Y       | X                | 16.9   | 9.5  | 23.8 |
> | X-Decoder (B)      | UniCL     | Y     | Y    | Y       | Y                | 21.1   | 11.7 | 27.2 |
> | SEEM (davit-d3)    | X-Decoder | Y     | Y    | X       | X                | 16.7   | 11.6 | 23.5 |
> | SEEM (samvit-B)    | SAM       | Y     | Y    | X       | X                | 13.3   | 9.5  | 18.9 |
>
> We will incorporate this discussion in our revision.
>
> **Q3: Comparisons of model parameters and calculations.**
>
> Thanks for pointing out this. In this work, our goal is to develop a versatile and universal image segmentation interface and thus we mainly focused on the comparisons with previous work in terms of task breadth and performance. Regarding the model parameters and calculations, we did list the rough size of backbones used in different methods (tiny (T), base (B), and large (L)), but failed to find a well-established way from the prior arts to analyze the computation of each generalist model, considering it is task-dependent while each model is used for different sets of tasks. Moreover, we faced the difficulty that the weight and inference code are not necessarily released for each specific task. That being said, we are not able to provide a thorough analysis of calculations for now but will definitely try our best to find a common-grounded way to solidly study this aspect.

---

> ### Author Response · Authors · 2023-08-14
>
> Hi reviewer cQYZ, thanks for your reviews again! And hope the rebuttal material solve your confusion.
>
> It would be really appreciated if you are willing to give any additional comment, and feel free to drop any question if you have any confusion on the rebuttal material : )

---

> > ### Comment · Reviewer_cQYZ · 2023-08-15
> >
> > Thanks to the authors for the response. My concerns have been addressed and I will keep my positive rating.

---

> > > ### Author Response · Authors · 2023-08-20
> > >
> > > Hi reviewer  cQYZ,
> > >
> > > Thanks so much for your review and positive comments! It would be really appreciated if you could **confirm your final recommendation rating before the final deadline on Aug 21st 1 pm EDT** : )
> > >
> > > Thanks a lot!

---

### Author Rebuttal · Authors · 2023-08-10

First of all, **we thank all reviewers for their valuable comments and suggestions!**

We sincerely appreciate all reviewers’ time and efforts in reviewing our paper. We are glad to find that reviewers generally recognized our contributions:

**Model.** A strong generalization ability that is able to apply to more scenarios (cQYZ). Enabling open vocabulary segmentation in addition to interactive segmentation (cQYZ). Highlight the trend towards more flexible segmentation models (R6GZ). The proposed method is technical novel (R6GZ). Innovative and highly effective (x17t). Unifying several segmentation tasks (gMmy).

**Experiments.** Superior performance across different types of segmentation tasks (cQYZ). The single pre-trained SEEM model achieves competitive performance across all segmentation tasks (R6GZ). The experimental results are solid (x17t). Impressive results on most tasks (gMmy).

**Writing.** The paper is well-presented, showcasing clear and concise writing (x17t). Reasonably well written (gMmy).

**Two Additional Contributions:**

* We add multi-instance visual prompt training and inference for interactive segmentation (Refer to Fig.1 in Rebuttal PDF). That enables query multiple objects in a single forward pass.

* We use SAM ViT backbone as the pretrained checkpoint to "scale up" our training dataset.

|          | Backbone | Multi-Queries in Single Inference | 20-NoC@50 | 20-NoC@85 | 20-NoC90 | 1-IoU |
|----------|----------|------------------------------------|-----------|-----------|----------|-------|
| SAM (B)  | SAM-B    | X                                  | /         | 3.28      | 4.12     | /     |
| SAM (L)  | SAM-L    | X                                  | /         | 2.60      | 3.12     | /     |
| SAM (H)  | SAM-H    | X                                  | /         | 2.55      | 3.11     | /     |
| SEEM (B) | Davit-d3 | X                                  | /         | 2.93      | 3.79     | /     |
| SEEM (L) | Davit-d5 | X                                  | /         | 2.77      | 3.61     | /     |
| SEEM (B) | Davit-d3 | Y                                  | 1.37      | 2.71      | 3.43     | 87.2  |
| SEEM (L) | Davit-d5 | Y                                  | 1.38      | 2.70      | 3.47     | 88.5  |
| SEEM (B) | SAM-B    | Y                                  | 1.29      | 2.53      | 3.21     | 86.2  |
| SEEM (L) | SAM-L    | Y                                  | 1.30      | 2.42      | 2.97     | 84.5  |

We found both techniques improves the performance with a good margin.

**New Experiments:**

* Open-Vocabulary Generic Segmentation on ADE20K dataset in comparison with X-Decoder baselines. (Reviewer cQYZ, R6GZ)

* "Scaling up" with SAM pretrained checkpoint. (Reviewer x17t)

* Enable multiple interactive query in a single forward pass.

* Ablation Study on gradually removing prompt type. (Reviewer x17t)

We carefully read all comments and attempted to address the concerns with comprehensive responses, and hope our responses could answer the questions and resolve any concerns regarding our work. Again, we thank all reviewers again for their efforts and constructive comments!

---

### Decision · Program_Chairs · 2023-09-21

**Decision:**

Accept (poster)

**Comment:**

All reviewers reached a consensus to accept the paper. The authors are encouraged to incorporate all the feedback and suggestions from the reviewers into the final version of the manuscript.